# Free and Transient Vibration Analysis of Sandwich Piezoelectric Laminated Beam with General Boundary Conditions

**DOI:** 10.3390/ma19010136

**Published:** 2025-12-30

**Authors:** Xiaoshuai Zhang, Wei Fu, Zixin Ning, Ningze Sun, Yang Li, Ziyuan Yang, Sen Jiu

**Affiliations:** 1China Ordnance Industry Group Aviation Ammunition Research Institute Co., Ltd., China North Industries Group Corporation Limited, Harbin 150030, China; 15545593315@163.com; 2College of Mechanical and Electrical Engineering, Harbin Engineering University, Harbin 150001, China; 13521178147@163.com (Z.N.); 18045623837@163.com (Y.L.); yangziyuan1108@163.com (Z.Y.); js16637718196@163.com (S.J.); 3College of Mechanical & Energy Engineering, Beijing University of Technology, Beijing 100124, China; snz2022@emails.bjut.edu.cn

**Keywords:** sandwich piezoelectric laminated beam, free vibration, transient response, method of reverberation ray matrix, general boundary conditions

## Abstract

This study comprehensively analyzes the free vibration and transient response for a sandwich piezoelectric laminated beam with elastic boundaries in a thermal environment. Quasi-3D shear deformation beam theory (Q3DBT) and Hamilton’s principle are used to obtain the thermo-electro-mechanical coupling equations, and the method of reverberation-ray matrix (MRRM) is utilized to integrate the phase and scattering relationship of the structure in a unified approach. Specifically, the scattering relationship established by the Mixed Rigid-Rod Model (MRRM) via dual coordinate systems describes the general dynamic model of the beam using generalized displacements and generalized forces at the two endpoints. This analytical solution is compared with the finite element numerical results based on Solid5 and Solid45 elements. The similarity of this approach lies in the fact that solid elements can account for the Poisson effect of thick beams, while the difference is that solid elements have a certain width; here, the error is minimized by adopting a single-element division in the width direction. Comparison of the numerical results under different geometric parameters and boundary conditions with the simulation software proves that MRRM has good accuracy and stability in analyzing the dynamic performance of sandwich piezoelectric laminated beams. On this basis, a spring-supported boundary technology is introduced to expand the flexibility of classical boundary conditions, and a detailed parameterization study is conducted on the material properties of the base layer, including the material parameters, geometric property, and the external temperature. The study in this article provides many new results for sandwich-type piezoelectric laminated structures to help further research.

## 1. Introduction

Smart drivers are gradually being used in various fields due to their excellent performance and high efficiency. Piezoelectric ceramics—which, as smart materials with positive and negative piezoelectric effects, are being widely used in new actuators such as ultrasonic motors—are an example of this. The stator of the motor is formed by two layers of piezoelectric ceramic sheets pasted on the base in the middle. These drives rely on vibration modes and resonance to output power [1,2,3]. The sandwich composite structure has better design characteristics than ordinary composite beams in terms of material and geometric parameters of the base layer, and the laminate scheme combined with composite materials is worthy of a series of parameter studies. In practical applications, such smart actuators are generally installed on high-speed aircraft to drive key parts such as the wing and rudder, which may carry large mechanical–thermal loads or even shock loads to form large transient accelerations and cause structural instability. Therefore, conducting a unified analysis of the dynamic performance for the sandwich piezoelectric structure in a thermal environment and finding better design parameters are necessary [4,5].

To make the vibration generated by the piezoelectric sheet of linear piezoelectric actuators better act on the motor mover, the sandwich structure is generally designed with a slenderness ratio of four or larger; such a structure can be regarded as a two-dimensional beam model. The related deformation theories on beams have been well developed, such as Classical Lamination Theory (CLT) [6,7], First-Order Shear Deformation Theory (FSDT) [8,9,10], and High-Order Shear Deformation Theory (HSDT) [11,12,13,14]. The dynamic analysis of piezoelectric beams in the past used FSDT more often and neglected the thickness stretch effect. Their models include bimorph beams [15,16], sandwich beams [17,18,19], micro beams [20,21], or functionally graded piezoelectric (FGP) beams [22,23,24]. Part of the research using HSDT mainly focuses on free vibration and buckling analysis. Tornabene et al. present a higher-order thermo-electro-elastic formulation for analyzing smart laminated shells [25]. Zenkour and Aljadani [26] studied the electro-mechanical buckling response of an FGP plate, and they numerically analyzed its external voltage, geometry, and material index. Arefi [27] used four HSDTs and the differential quadrature method (DQM) on the basis of nonlocal elastic theory to analyze the free vibration of the sandwich micro Functionally Graded (FG) curved beam and conducted parametric studies on different shear shape functions and volume fractions.

The thickness stretch effect of medium-thick and thick beams will have a certain effect on free vibration and transient response. Quasi-3D shear deformation beam theory (Q3DBT) combines multiple shear shape functions and proposes kinematic equations with thickness stretch effect in the z direction [28,29,30]. Thuc P.Vo and Thai [31] established finite element models and used Q3DBT to study the geometric parameters, frequency mode characteristics, and buckling load of FG materials. Karamanlı [32] provided various static mechanical properties of FG sandwich beams based on Q3DBT and symmetric smoothed particle hydrodynamics method. Polit and Anant [33] used Q3DBT with a sinusoidal shape function to study FG porous micro curved beams under different porosity and geometric parameters, and they analyzed the deflection and stress of the beam. On the basis of the same shape function, Arani et al. [34] derived the governing equations of piezoelectric micro plates embedded in composite polymers with nonlocal theory, and they obtained the effect of various material and geometry parameters on the wave propagation of micro plates through analytical solutions. Sayyad and Ghugal [35] conducted a bending analysis of sandwich curved beams with FG skins pasted on the top and bottom of the base beam. They used Navier’s solution method to solve and provide the results under different curvatures and power laws. The abovementioned studies show that many works have been conducted on Functionally Graded Material (FGM) laminated beams, while piezoelectric laminated beams are concentrated on static analysis and lack a unified dynamic characteristic research. Research on the Poisson effect (thickness stretching effect) and the transient response in a thermal environment of sandwich-type piezoelectric laminated beams is also lacking.

Dynamic performance analysis derives and solves the governing differential equations of the model. The related solving methods mainly include the finite element method (FEM) [36,37,38], transfer matrix method (MTM) [39,40], spectrum element method (SEM) [41,42,43], DQM [44,45,46], Ritz method [47], and Galerkin method [48]. FEM, as a general calculation program, has powerful multi-field coupling solving capabilities, including static response, modal, and steady state. However, it will encounter low efficiency when solving high-precision problems because it is an approximate calculation method. The MTM is more suitable for solving dynamic problems of laminated structures and has higher efficiency, but it is prone to unstable numerical problems because of a single transfer matrix. SEM is based on the stiffness matrix and has a unified column formula, which is suitable for the wave analysis of composite structures. Its drawback is that the problem of numerical instability may occur when inverting the spectral stiffness matrix. The method of reverberation-ray matrix (MRRM) combines the advantages of MTM and the SEM to avoid the defects in the abovementioned solution method. The formulas are more uniform and contain a clear physical meaning of the scattering matrix by classifying the traveling waves into the departing and arriving waves. As a result, the problem of large indexes in MTM is effectively avoided. Miao et al. [49] extended MRRM to transient analysis and studied the response of composite beams under different external loads. The comparison with FEM proves that MRRM is a simple and effective dynamic solution method. Guo et al. [50] used MRRM to propose a numerical analysis for a multilayer piezoelectric structure, and the results proved that MRRM has good numerical stability and is not affected by the geometric parameters of the structure. Chen [51] used a recursive algorithm of MRRM to study the vibration characteristics of classic beams, and the results proved that MRRM has a stable value when calculating high-order frequencies. However, to the best of the authors’ knowledge, no unified study has been conducted on the dynamic response of MRRM to sandwich piezoelectric laminated beams.

The previous studies of sandwich laminated beams have a narrow focus on the boundary condition and mostly focused on clamped, simple support, and free. Some works proposed a general method for composite laminated and FG beams by using a series of springs to simulate boundary forces. Zhao and Wang [52] studied the free vibration of an elliptical cylindrical shell under a combination of general and classical boundary conditions. Su and Jin [53] used FSDT to analyze free vibration and transient vibration of a single-layer FG curved beam under general boundary conditions. These studies indicate that arbitrary boundary conditions can be achieved by adjusting the stiffness coefficient of the spring. Therefore, studying different dynamic responses for sandwich piezoelectric laminated beams under general boundary conditions is necessary. MRRM, as a solution method in the frequency domain, can introduce a spring boundary corresponding to the displacement field through the phase and scattering relationship under dual coordinates. As mentioned above, this study takes the sandwich piezoelectric laminated beam as the research model. Several dynamic response results under spring boundaries with arbitrary stiffness coefficients are obtained and compared with FEM software (Ansys 18.2) on the basis of Q3DBT combined with Reddy’s shear shape functions and the MRRM. The effect of external thermal load is also considered. Parametric studies are intended to improve and enrich the unity of MRRM analysis of this type of structure, including the material parameters of the base layer, the overall geometric properties, the external temperature rise, and the response characteristics under various elastic boundary combinations.

This paper addresses the thermal–structural coupling and complex boundary support issues faced by the sandwich piezoelectric composite beams applied in intelligent drive systems under actual working conditions. By using the quasi-3-dimensional shear deformation beam theory (Q3DBT) that takes into account the Poisson effect and Hamilton’s principle, the thermal-electro-structural coupling control equations of the structure are derived. Subsequently, the back-propagation ray matrix method (MRRM) is introduced to integrate the phase and scattering relationships of wave propagation within a unified framework, achieving accurate and efficient analysis of the free vibration and transient response of sandwich piezoelectric beams with general elastic boundary conditions. The numerical examples are used to verify the accuracy of the method, and the influence laws of matrix material parameters, geometric characteristics, layup method, and temperature changes on the dynamic performance of the structure are systematically explored. The aim is to provide a theoretical basis and parameterization guidance for the design optimization and engineering application of such structures.

## 2. Geometric and Material Equation

### 2.1. Research Model

As illustrated in Figure 1, the sandwich laminated beam comprises two layers of piezoelectric sheets and a base layer. The length of the beam is *L*, the thickness of the upper and lower piezoelectric layers is *h_p_*, and the thickness of the middle substrate layer is *h_m_*. Five sets of springs are installed on the end faces of the beam in order to accurately simulate various mechanical boundary conditions. In this configuration, 
$\left\{k_{u_{0}},k_{w_{0}},k_{w_{z}}\right\}$
 denote the spring stiffness in the *x* and *z* directions, while 
$\left\{k_{\theta_{xb}},k_{\theta_{xs}}\right\}$
 represent the rotation spring stiffness. Moreover, consideration is given to one particular set of constraints, *k_Φ_*, which is associated with electric displacement. The piezoelectric term is incorporated in the derivation of the subsequent equations, and the base layer can be adjusted arbitrarily according to the actual situation.

### 2.2. Geometric Equation

According to the Q3DBT, the geometric equation of any point on the midplane of the beam can be written as [31]
(1)
$$u(x,z,t)=u_{0}(x,t)+z\theta_{xb}(x,t)+f(z)\theta_{xs}(x,t)$$

(2)
$$w(x,z,t)=w_{0}(x,t)+g(z)w_{z}(x,t)$$

(3)
$$\theta_{xb}(x,t)=-\frac{\partial w_{0}(x,t)}{\partial x}$$

where *g*(*z*) and *w_z_* are used to investigate the effect of thickness stretching of the piezoelectric laminated beam on the dynamic characteristics. *f*(*z*) is the original function of *g*(*z*), and both are shape functions to describe the transverse shear along the thickness. Obviously, FSDT and CBT can be obtained by setting *f*(*z*) = *z*, *g*(*z*) = 0 and *f*(*z*) = 0, *g*(*z*) = 0. The shape functions in this study are set based on Reddy’s beam theory as *f*(*z*) = *z* − 4*z*^3^/3*h*^2^, *g*(*z*) = 1 − 4*z*^2^/*h*^2^. In addition, the electric potential variables associated with the piezoelectric material are given as [33]
(4)
$$\phi (x,z)=-\cos(\alpha z)\phi (x)+\frac{2z\varphi_{0}}{h}$$

where *α* = *π*/*h*, *Φ*_0_ is initial external voltage.

Mechanical strains and electric field can be obtained from Equations (1)–(4) and written in the form of strain components as
(5)
$$\varepsilon_{xx}=\frac{\partial u}{\partial x}=\left[\varepsilon_{xx}^{0}+z\varepsilon_{xx}^{xb}+f(z)\varepsilon_{xx}^{xs}\right]=\frac{\partial u_{0}}{\partial x}+z\frac{\partial \theta_{xb}}{\partial x}+f(z)\frac{\partial \theta_{xs}}{\partial x}$$

(6)
$$\varepsilon_{zz}=\frac{\partial w}{\partial z}=g^{\prime }(z)\varepsilon_{zz}^{0}=g^{\prime }(z)w_{z}$$

(7)
$$\gamma_{xz}=\frac{\partial u}{\partial z}+\frac{\partial w}{\partial x}=g(z)\gamma_{xz}^{0}=g(z)\left(\frac{\partial w_{z}}{\partial x}+\theta_{xs}\right)$$

(8)
$$E_{x}=-\frac{\partial \phi }{\partial x}=\cos(\alpha z)\frac{\partial \phi }{\partial x}$$

(9)
$$E_{z}=-\frac{\partial \phi }{\partial z}=-\alpha \sin(\alpha z)\phi -\frac{2\varphi_{0}}{h}$$


Considering that a general piezoelectric sheet is an orthotropic material, the constitutive equations considering Poisson’s ratio effect can be expressed as [33,54]
(10)
$$\left[\begin{array}{c} \sigma_{xx}\\ \sigma_{yy}\\ \sigma_{zz}\\ \tau_{yz}\\ \tau_{xz}\\ \tau_{xy} \end{array}\right]=\left[\begin{array}{cccccc} {\bar{c}}_{11} & {\bar{c}}_{12} & {\bar{c}}_{13} & 0 & 0 & 0\\ {\bar{c}}_{12} & {\bar{c}}_{22} & {\bar{c}}_{23} & 0 & 0 & 0\\ {\bar{c}}_{13} & {\bar{c}}_{23} & {\bar{c}}_{33} & 0 & 0 & 0\\ 0 & 0 & 0 & {\bar{c}}_{44} & 0 & 0\\ 0 & 0 & 0 & 0 & {\bar{c}}_{55} & 0\\ 0 & 0 & 0 & 0 & 0 & {\bar{c}}_{66} \end{array}\right]\left[\begin{array}{c} \varepsilon_{xx}\\ \varepsilon_{yy}\\ \varepsilon_{zz}\\ \gamma_{yz}\\ \gamma_{xz}\\ \gamma_{xy} \end{array}\right]-\left[\begin{array}{ccc} 0 & 0 & {\bar{e}}_{31}\\ 0 & 0 & {\bar{e}}_{31}\\ 0 & 0 & {\bar{e}}_{33}\\ 0 & {\bar{e}}_{15} & 0\\ {\bar{e}}_{15} & 0 & 0\\ 0 & 0 & 0 \end{array}\right]\left[\begin{array}{c} E_{x}\\ E_{y}\\ E_{z} \end{array}\right]$$

(11)
$$\left[\begin{array}{c} D_{x}\\ D_{y}\\ D_{z} \end{array}\right]=\left[\begin{array}{cccccc} 0 & 0 & 0 & 0 & {\bar{e}}_{15} & 0\\ 0 & 0 & 0 & {\bar{e}}_{15} & 0 & 0\\ {\bar{e}}_{31} & {\bar{e}}_{32} & {\bar{e}}_{33} & 0 & 0 & 0 \end{array}\right]\left[\begin{array}{c} \varepsilon_{xx}\\ \varepsilon_{yy}\\ \varepsilon_{zz}\\ \gamma_{yz}\\ \gamma_{xz}\\ \gamma_{xy} \end{array}\right]+\left[\begin{array}{ccc} {\bar{s}}_{11} & 0 & 0\\ 0 & {\bar{s}}_{11} & 0\\ 0 & 0 & {\bar{s}}_{33} \end{array}\right]\left[\begin{array}{c} E_{x}\\ E_{y}\\ E_{z} \end{array}\right]$$


Equations (10) and (11) represent the mechanical stress and electric displacement of the piezoelectric layer, respectively.
(12)
$$\left[\begin{array}{c} \sigma_{xx}^{T(k)}\\ \sigma_{yy}^{T(k)}\\ \sigma_{zz}^{T(k)}\\ \tau_{xy}^{\left(k\right)} \end{array}\right]=\Delta T\left[\begin{array}{cccc} {\bar{c}}_{11} & {\bar{c}}_{12} & {\bar{c}}_{13} & 0\\ {\bar{c}}_{12} & {\bar{c}}_{22} & {\bar{c}}_{23} & 0\\ {\bar{c}}_{13} & {\bar{c}}_{23} & {\bar{c}}_{33} & 0\\ 0 & 0 & 0 & {\bar{c}}_{66} \end{array}\right]\left[\begin{array}{c} \alpha_{xx}^{\left(k\right)}\\ \alpha_{yy}^{\left(k\right)}\\ \alpha_{zz}^{\left(k\right)}\\ \alpha_{xy}^{\left(k\right)} \end{array}\right]$$

(13)
$$D_{z}^{T}={\bar{p}}_{3}\Delta T$$


Equations (12) and (13) can be regarded as the thermal stress caused by thermal expansion *α*_ii_ and pyroelectric coefficient 
${\bar{p}}_{3}$
 when a temperature rise occurs in the environment. *σ*_zz_ and *ε*_zz_ are the terms related to the Poisson effect (thickness stretching effect). 
${\bar{c}}_{ij}$
, 
${\bar{e}}_{ij}$
, and 
${\bar{s}}_{ij}$
 represent the transformed elastic, piezoelectric, and dielectric coefficients. 
D
 represents the thermal displacement. We can simply express Equations (12) and (13) as [55]
(14)
$${\bar{c}}_{11}=c_{11}-\frac{c_{13}^{2}}{c_{33}},\;{\bar{c}}_{13}=c_{13},\;{\bar{c}}_{55}=c_{55}$$

(15)
$${\bar{e}}_{31}=e_{31}-e_{33}\frac{c_{13}}{c_{33}},\;{\bar{e}}_{33}=e_{33},\;{\bar{e}}_{15}=e_{15}$$

(16)
$${\bar{s}}_{11}=s_{11},\;{\bar{s}}_{33}=s_{33}+\frac{e_{33}^{2}}{c_{33}}$$

(17)
$${\bar{p}}_{3}=p_{3}-\frac{e_{33}p_{3}}{c_{33}}$$

where matrixes ***e***, ***E***, and ***s*** should be neglected when the corresponding *k*-th layer is the base layer. The base layer is the central metal layer of the sandwich beam, which acts as the driven structure with piezoelectric patches attached to its surfaces for actuation. The coefficients in the abovementioned matrixes are defined as
(18)
$$\sigma^{\left(k\right)}={\bar{c}}^{\left(k\right)}\varepsilon^{\left(k\right)}-{\bar{e}}^{\left(k\right)}E^{\left(k\right)}$$

(19)
$$D^{\left(k\right)}={\bar{e}}^{\left(k\right)}\varepsilon^{\left(k\right)}+{\bar{s}}^{\left(k\right)}E^{\left(k\right)}$$


Here, *c*_11_, *c*_33_, *c*_13_ are the elastic constants and *e*_31_, *e*_33_ are the piezoelectric constants to determine the bending and extension modes, respectively. A piezoelectric laminated beam can be regarded as a simplified plate structure. 
$\left\{\sigma_{yy},\tau_{xy},\tau_{yz}\right\}$
, *D_y_*, and 
$\left\{\sigma_{yy}^{T(k)},\tau_{xy}^{T(k)}\right\}$
 in Equations (10)–(12) can be negligible when the width-to-length ratio is very small. By integrating the mechanical stresses from the remaining stress terms in the z direction, we can obtain
(20)
$$N_{xx}={\tilde{A}}_{11}\frac{\partial u_{0}}{\partial x}+{\tilde{B}}_{11}\frac{\partial \theta_{xb}}{\partial x}+{\tilde{M}}_{11}\frac{\partial \theta_{xs}}{\partial x}+J_{13}w_{z}+E_{31}\Phi$$

(21)
$$M_{xxb}={\tilde{B}}_{11}\frac{\partial u_{0}}{\partial x}+{\tilde{D}}_{11}\frac{\partial \theta_{xb}}{\partial x}+{\tilde{N}}_{11}\frac{\partial \theta_{xs}}{\partial x}+K_{13}w_{z}+F_{31}\Phi$$

(22)
$$M_{xxs}={\tilde{M}}_{11}\frac{\partial u_{0}}{\partial x}+{\tilde{N}}_{11}\frac{\partial \theta_{xb}}{\partial x}+{\tilde{O}}_{11}\frac{\partial \theta_{xs}}{\partial x}+L_{13}w_{z}+E_{31}^{f}\Phi$$

(23)
$$\begin{array}{l} N_{zz}=\sum_{k=1}^{N}\int_{z^{\left(k\right)}}^{z^{\left(k+1\right)}}g^{\prime }(z)\sigma_{zz}^{\left(k\right)}dz\\ \;\;\;\;\;\;=J_{13}\left(\frac{\partial u_{0}}{\partial x}\right)+K_{13}\frac{\partial \theta_{xb}}{\partial x}+L_{13}\frac{\partial \theta_{xs}}{\partial x}+G_{33}w_{z}+E_{33}^{g^{\prime }}\Phi \end{array}$$

(24)
$$D_{x}=E_{31}\frac{\partial u_{0}}{\partial x}+F_{31}\frac{\partial \theta_{xb}}{\partial x}+E_{31}^{f}\frac{\partial \theta_{xs}}{\partial x}+E_{33}^{g\prime }w_{z}-X_{33}\phi$$

(25)
$$Q_{xz}=O_{55}^{S}\left(\frac{\partial w_{z}}{\partial x}+\theta_{xs}\right)-E_{15}^{g}\frac{\partial \Phi }{\partial x}$$

(26)
$$D_{z}=E_{15}^{g}\left(\theta_{xs}+\frac{\partial w_{z}}{\partial x}\right)+X_{11}\frac{\partial \phi }{\partial x}$$


Similarly, every group of stiffness coefficients can be written as
(27)
$$\left\{A_{11},B_{11},D_{11},M_{11},N_{11},O_{11}\right\}=\sum_{k=1}^{N}\int_{z^{\left(k\right)}}^{z^{\left(k+1\right)}}{\bar{c}}_{11}^{\left(k\right)}\left\{1,z,z^{2},f(z),zf(z),f^{2}(z)\right\}dz$$

(28)
$$\left\{J_{13},K_{13},L_{13}\right\}=\sum_{k=1}^{N}\int_{z^{\left(k\right)}}^{z^{\left(k+1\right)}}{\bar{c}}_{13}^{\left(k\right)}\left\{g^{\prime }(z),zg^{\prime }(z),f(z)g^{\prime }(z)\right\}dz$$

(29)
$$G_{33}=\sum_{k=1}^{N}\int_{z^{\left(k\right)}}^{z^{\left(k+1\right)}}{\bar{c}}_{33}^{\left(k\right)}{g^{\prime }}^{2}(z)dz$$

(30)
$$\left\{E_{31},F_{31},E_{31}^{f}\right\}=\sum_{k=1}^{N}\int_{z^{\left(k\right)}}^{z^{\left(k+1\right)}}{\bar{e}}_{31}^{\left(k\right)}\left\{1,z,f(z)\right\}\alpha \sin(\alpha z)dz$$

(31)
$${E_{33}}^{g^{\prime }}=\sum_{k=1}^{N}\int_{z^{\left(k\right)}}^{z^{\left(k+1\right)}}{\bar{e}}_{33}^{\left(k\right)}{g^{\prime }}^{2}(z)\alpha \sin(\alpha z)dz,\;{E_{15}}^{g}=\sum_{k=1}^{N}\int_{z^{\left(k\right)}}^{z^{\left(k+1\right)}}{\bar{e}}_{15}^{\left(k\right)}g(z)\cos(\alpha z)dz$$

(32)
$$X_{11}=\sum_{k=1}^{N}\int_{z^{\left(k\right)}}^{z^{\left(k+1\right)}}{\bar{s}}_{11}^{\left(k\right)}\cos^{2}(\alpha z)dz,\;X_{33}=\sum_{k=1}^{N}\int_{z^{\left(k\right)}}^{z^{\left(k+1\right)}}{\bar{s}}_{33}^{\left(k\right)}{\left[\alpha \sin(\alpha z)\right]}^{2}dz$$


### 2.3. Equilibrium Governing Equations

It is possible to obtain the governing differential equations of the sandwich laminated beam under a thermo-electro coupling field by employing Hamilton’s principle [12].
(33)
$$\int_{t_{1}}^{t_{2}}\int_{V}\left(\delta U_{M}+\delta U_{E}-\delta K\right)dVdt=0$$

where *δU_M_*, *δU_E_*, and *δK* represent virtual strain energy, external force potential energy, and kinetic energy, respectively. In this paper, the beam model is developed based on the Quasi-3D shear deformation theory. In this model, the axial and transverse deformations (denoted as u and w, respectively) are considered, while the deformation along the width direction is neglected. Consequently, the integration over the beam width is omitted. δ*U_M_* is generally written as
(34)
$$U_{S}=\int_{0}^{L}\int_{-h/2}^{h/2}\left(\sigma_{xx}\varepsilon_{xx}+\tau_{xz}\gamma_{xz}+\sigma_{zz}\varepsilon_{zz}-D_{x}E_{x}-D_{z}E_{z}\right)dzdx$$

(35)
$$\delta U_{S}=\int_{0}^{L}\left\{\begin{array}{l} N_{xx}\frac{\partial \delta u_{0}}{\partial x}+M_{xxb}\frac{\partial \delta \theta_{xb}}{\partial x}+Q_{xz}(\frac{\partial \delta w_{z}}{\partial x}+\delta \theta_{xs})+N_{zz}\delta w_{z}\\ \int_{-h/2}^{h/2}(-D_{x}\cos(\alpha z)\frac{\partial \delta \phi }{\partial x}-D_{z}\alpha \sin(\alpha z)\delta \phi )dz \end{array}\right\}dx$$


External forces include thermal and piezoelectric stresses related to temperature rise ∆*T* and electric potential *φ*_0_. In the beam structure studied in this work, only the stress occurring in the *x*-*z* plane is considered. The strain caused by external forces can be written in the form of axial strain as [56]
(36)
$$\varepsilon_{xx}^{E}=\frac{1}{2}{\left(\frac{\partial w}{\partial x}\right)}^{2}=\frac{1}{2}{\left[\frac{\partial w_{0}}{\partial x}+g(z)\left(\frac{\partial w_{z}}{\partial x}\right)\right]}^{2}$$


Thus, the virtual thermo-electro potential energy *δU_E_* should be written as
(37)
$$\begin{array}{l} \delta U_{E}=\int_{0}^{L}\int_{-h/2}^{h/2}\left(\sigma_{xx}^{T(k)}-2{\bar{e}}_{31}\varphi_{0}\right)\varepsilon_{xx}^{E}\\ \;\;\;\;\;\;=\int_{0}^{L}\left\{\begin{array}{l} \left(N_{xx}^{T}+N_{xx}^{E}\right)\frac{\partial w_{0}}{\partial x}\delta \frac{\partial w_{0}}{\partial x}+\left(N_{xx}^{gT}+N_{xx}^{gE}\right)\frac{\partial w_{0}}{\partial x}\delta \frac{\partial w_{z}}{\partial x}+\\ \left(N_{xx}^{gT}+N_{xx}^{gE}\right)\frac{\partial w_{z}}{\partial x}\delta \frac{\partial w_{0}}{\partial x}+\left(N_{xx}^{g^{2}T}+N_{xx}^{g^{2}E}\right)\frac{\partial w_{z}}{\partial x}\delta \frac{\partial w_{z}}{\partial x} \end{array}\right\} \end{array}$$

where 
$\left\{N_{xx}^{T},N_{xx}^{E}\right\}$
, 
$\left\{N_{xx}^{gT},N_{xx}^{gE}\right\},$
 and 
$\left\{N_{xx}^{g^{2}T},N_{xx}^{g^{2}E}\right\}$
 can be obtained in the same way as Equations (20)–(24):
(38)
$$\left\{N_{xx}^{T},N_{xx}^{gT},N_{xx}^{g^{2}T}\right\}=\sum_{k=1}^{N}\int_{z^{\left(k\right)}}^{z^{\left(k+1\right)}}\sigma_{xx}^{T(k)}\left\{1,g,g^{2}\right\}dz$$

(39)
$$\left\{N_{xx}^{E},N_{xx}^{gE},N_{xx}^{g^{2}E}\right\}=\sum_{k=1}^{N}\int_{z^{\left(k\right)}}^{z^{\left(k+1\right)}}\left(-2e_{31}\phi_{0}\right)\left\{1,g,g^{2}\right\}dz$$

(40)
$$\sigma_{xx}^{T\left(k\right)}={\tilde{c}}_{11}^{\left(k\right)}\varepsilon_{xx}^{T\left(k\right)}+{\tilde{c}}_{13}^{\left(k\right)}\varepsilon_{zz}^{T\left(k\right)}$$


The kinetic energy of the laminated beam can be expressed as
(41)
$$K=\int_{0}^{L}\int_{-h/2}^{h/2}\rho^{\left(k\right)}\left(\frac{\partial^{2}u}{\partial t^{2}}+\frac{\partial^{2}w}{\partial t^{2}}\right)dzdx$$


By substituting Equations (1)–(3) into Equation (41), we can get
(42)
$$\begin{array}{l} \delta K=-\int_{0}^{L}I_{0}\left({\ddot{u}}_{0}\delta u_{0}+{\ddot{w}}_{0}\delta w_{0}\right)-I_{1}\left(\frac{\partial {\ddot{w}}_{0}}{\partial x}\delta u_{0}+{\ddot{u}}_{0}\frac{\partial \delta w_{0}}{\partial x}\right)+I_{2}\left(\frac{\partial {\ddot{w}}_{0}}{\partial x}\frac{\partial \delta w_{0}}{\partial x}\right)\\ \;\;\;\;\;\;+I_{3}\left({\ddot{\theta }}_{xs}\delta u_{0}+{\ddot{u}}_{0}\delta \theta_{xs}\right)-I_{4}\left({\ddot{\theta }}_{xs}\frac{\partial \delta w_{0}}{\partial x}+\frac{\partial {\ddot{w}}_{0}}{\partial x}\delta \theta_{xs}\right)+I_{5}{\ddot{\theta }}_{xs}\delta \theta_{xs}\\ \;\;\;\;\;\;+I_{6}\left({\ddot{w}}_{0}\delta w_{z}+{\ddot{w}}_{z}\delta w_{0}\right)+I_{7}{\ddot{w}}_{z}^{2}\delta w_{z} \end{array}$$

where “⋅⋅” denotes the second derivative of time. *Ii* means the moment of inertia of the laminated beam and is calculated by integrating the density *ρ* of each layer in the *z* direction as
(43)
$$\left\{I_{0},I_{1},I_{2},I_{3},I_{4},I_{5},I_{6},I_{7}\right\}=\sum_{k=1}^{N}\int_{z^{\left(k\right)}}^{z^{\left(k+1\right)}}\rho^{\left(k\right)}\left\{1,z,z^{2},f,zf,f^{2},g,g^{2}\right\}dz$$


By substituting Equations (34), (35), (37), and (41) into Equation (33), the following equilibrium governing equations can be obtained by performing partial integration of all partial derivatives:
(44)
$$\begin{array}{l} 0=\int_{0}^{L}\left\{\begin{array}{l} \left(-\frac{\partial N_{xx}}{\partial x}+I_{0}{\ddot{u}}_{0}+I_{1}{\ddot{\theta }}_{xb}+I_{3}{\ddot{\theta }}_{xs}\right)\delta u_{0}\\ \left(-\frac{\partial Q_{xxb}}{\partial x}+I_{0}{\ddot{w}}_{0}+I_{6}{\ddot{w}}_{z}\right)\delta w_{0}\\ \left(-\frac{\partial M_{xxs}}{\partial x}+Q_{xz}+I_{3}{\ddot{u}}_{0}+I_{5}{\ddot{\theta }}_{xs}+I_{4}{\ddot{\theta }}_{xb}\right)\delta \theta_{xs}\\ \left(-\frac{\partial Q_{xxz}}{\partial x}+N_{zz}+I_{6}{\ddot{w}}_{0}+I_{7}{\ddot{w}}_{z}^{2}\right)\delta w_{z}\\ \int_{-h/2}^{h/2}\left(\frac{\partial D_{x}}{\partial x}\cos(\alpha z)+D_{z}\alpha \sin(\alpha z)\right)dz\delta \phi \end{array}\right\}dx\\ +{\left.N_{xx}\delta u_{0}+Q_{xxb}\delta w_{0}+M_{xxb}\delta \theta_{xb}+M_{xxs}\delta \theta_{xs}+Q_{xz}^{g}\delta w_{z}+\int_{-h/2}^{h/2}\left(D_{x}\cos(\alpha z)\delta \phi \right)dz\right|}_{0}^{L} \end{array}$$

where
(45)
$$Q_{xxb}=\frac{\partial M_{xxb}}{\partial x}-\left(N_{xx}^{E}+N_{xx}^{T}\right)\frac{\partial w_{0}}{\partial x}-\left(N_{xx}^{gE}+N_{xx}^{gT}\right)\frac{\partial w_{z}}{\partial x}-I_{1}{\ddot{u}}_{0}-I_{2}{\ddot{\theta }}_{xb}-I_{4}{\ddot{\theta }}_{xs}$$

(46)
$$Q_{xxz}=Q_{xz}^{g}-\left(N_{xx}^{gE}+N_{xx}^{gT}\right)\frac{\partial w_{0}}{\partial x}-\left(N_{xx}^{g^{2}E}+N_{xx}^{g^{2}T}\right)\frac{\partial w_{z}}{\partial x}$$


## 3. Solving Method

In this section, the MRRM will be introduced to analyze the free vibration and transient response of sandwich piezoelectric laminated beams with general boundary conditions. The solution to the above-mentioned dynamic characteristics, as an analysis method based on wave theory, can be attributed to the determination of the wave amplitude vectors by changing the boundary conditions, where the boundary conditions are the coupling relationship between the generalized force and displacement vectors. Furthermore, we consider an external incentive 
$g=g_{0}\delta (x-x_{0})$
 acting on the beam at *x*_0_, to perform the MRRM formulation. Equations (60)–(64) can be rewritten in the form of matrix multiplication as
(47)
$$\delta ={\left\{{\bar{u}}_{0},{\bar{w}}_{0},{\bar{\theta }}_{xb},{\bar{\theta }}_{xs},{\bar{w}}_{z},\bar{\phi }\right\}}^{T}=A_{\delta }P(-x)a+D_{\delta }P(x)d+q_{\delta }(x)$$

(48)
$$f={\left\{{\bar{N}}_{xx},{\bar{Q}}_{xxb},{\bar{M}}_{xxb},{\bar{M}}_{xxs},{\bar{Q}}_{xxz},\int_{-h/2}^{h/2}{\bar{D}}_{x}\cos(\alpha z)dz\right\}}^{T}=A_{f}P(-x)a+D_{f}P(x)d+q_{f}(x)$$

where
(49)
$$a={\left\{a_{1},a_{2},a_{3},a_{4},a_{5},a_{6}\right\}}^{T}$$

(50)
$$d={\left\{d_{1},d_{2},d_{3},d_{4},d_{5},d_{6}\right\}}^{T}$$

(51)
$$P(x)=diag\left(e^{-\lambda_{1}x},e^{-\lambda_{2}x},e^{-\lambda_{3}x},e^{-\lambda_{4}x},e^{-\lambda_{5}x},e^{-\lambda_{6}x}\right)$$

{*A*_δ_, *D*_δ_} and {*A*_f_, *D*_f_} are the generalized displacement and force coefficient matrixes corresponding to each group of wave numbers, which are given in Appendix C. ***g***_0_ is the amplitude of external excitation, and it is written as
(52)
$$g_{0}={\left\{{\tilde{N}}_{xx},{\tilde{Q}}_{xxb},{\tilde{M}}_{xxb},{\tilde{M}}_{xxs},{\tilde{Q}}_{xxz},\int_{-h/2}^{h/2}{\tilde{D}}_{x}\cos(\alpha z)dz\right\}}^{T}$$

where the inhomogeneous term in Equations (47) and (48) is calculated as
(53)
$$\left\{\begin{array}{c} q_{\delta }(x)\\ q_{f}(x) \end{array}\right\}=\left[\begin{array}{cc} A_{\delta }P(-x) & D_{\delta }P(x)\\ A_{f}P(-x) & D_{f}P(x) \end{array}\right]\int_{0}^{x}{\left[\begin{array}{cc} A_{\delta }P(-\xi ) & D_{\delta }P(\xi )\\ A_{f}P(-\xi ) & D_{f}P(\xi ) \end{array}\right]}^{-1}\left\{\begin{array}{c} 0\\ g_{0} \end{array}\right\}\delta (\xi -x_{0})d\xi$$


### 3.1. Wave Solutions

According to the fundamental analysis process of MRRM, wave solutions must undergo conversion to the frequency domain through the implementation of the Laplace transformation on the governing equations. The wave solution refers to a frequency-domain solution derived by subjecting the governing differential equations to a Laplace transform. Within composite laminated structures, elastic waves propagate across interlayers in the form of traveling waves. The expression can be interpreted as a wave mode propagating along the x-direction:
(54)
$$\bar{v}(x,s)=Ae^{\lambda x}V$$


This expression establishes an intuitive correlation between the mathematical homogeneous solution and the traveling wave propagation within the structure. Based on the structural delamination characteristics and interfacial connection conditions, all traveling wave expressions are coupled with the boundary conditions to formulate a unified matrix equation:
(55)
M(s)V=F(s)


Here, *M*(*s*) denotes the system matrix, which encapsulates the comprehensive effects of structural wave propagation characteristics and boundary constraints; *F*(*s*) represents the wave amplitude vector corresponding to external excitations.

Solving this matrix equation yields the wave amplitude *V* of each layer. Substituting *V* into the wave solution expression enables the derivation of the structural response in either the frequency domain or the time domain. The entire procedure explicitly maps the mathematical solution steps to the physical processes of wave propagation, reflection, and superposition.

Furthermore, Equation (44) is to be rewritten as
(56)
$$\left[\begin{array}{ccccc} L_{11} & L_{12} & L_{13} & L_{14} & L_{15}\\ L_{21} & L_{22} & L_{23} & L_{24} & L_{25}\\ L_{31} & L_{32} & L_{33} & L_{34} & L_{35}\\ L_{41} & L_{42} & L_{43} & L_{44} & L_{45}\\ L_{51} & L_{52} & L_{53} & L_{54} & L_{55} \end{array}\right]\left[\begin{array}{c} {\bar{u}}_{0}\\ {\bar{w}}_{0}\\ {\bar{\theta }}_{xs}\\ {\bar{w}}_{z}\\ \bar{\phi } \end{array}\right]=0$$

where the specific formulas of *L_ij_* can be found in Appendix A. The displacement term can generally be written in the form of wave number as
(57)
$$\left[\begin{array}{c} {\bar{u}}_{0}\\ {\bar{w}}_{0}\\ {\bar{\theta }}_{xs}\\ {\bar{w}}_{z}\\ \bar{\phi } \end{array}\right]=\left[\begin{array}{c} Ue^{\lambda x}\\ We^{\lambda x}\\ \vartheta e^{\lambda x}\\ W_{z}e^{\lambda x}\\ \psi e^{\lambda x} \end{array}\right]$$


In this equation, the symbol *λ* is used to indicate the so-called “characteristic wave number” in the *x* direction. Meanwhile, the symbols 
$\left\{U,W,\vartheta ,W_{z},\psi \right\}$
 correspond to the amplitude of displacement components in turn. The combination of Equations (56) and (57) allows for the subsequent equation to be derived:
(58)
$$\left[\begin{array}{ccccc} T_{11} & T_{12} & T_{13} & T_{14} & T_{15}\\ T_{21} & T_{22} & T_{23} & T_{24} & T_{25}\\ T_{31} & T_{32} & T_{33} & T_{34} & T_{35}\\ T_{41} & T_{42} & T_{43} & T_{44} & T_{45}\\ T_{51} & T_{52} & T_{53} & T_{54} & T_{55} \end{array}\right]\left[\begin{array}{c} U\\ W\\ \vartheta \\ W_{z}\\ \psi \end{array}\right]=0$$


The precise formulae of *T_ij_* can be located in Appendix B. It can be deduced that the subsequent 12-order equation and its concomitant six sets of wave numbers ±*λ_i_* (*i* = 1–6) can be obtained by assigning the matrix ***T*** the value of zero. It should be noted that bi is used to denote the corresponding coefficient, as illustrated below:
(59)
$$b_{12}\lambda^{12}+b_{10}\lambda^{10}+b_{8}\lambda^{8}+b_{6}\lambda^{6}+b_{4}\lambda^{4}+b_{2}\lambda^{2}+b_{0}=0$$

where bi is the coefficient of the characteristic wavenumber. Equation (58) can be solved by setting one term in 
${\left\{U,W,\vartheta ,W_{z},\psi \right\}}^{T}$
 equal to 1. For the convenience of calculation, the basic solution can be assumed as
(60)
$$\left\{\alpha_{i},\beta_{i},\gamma_{i},\xi_{i},1\right\}$$

where
(61)
$$\alpha_{i}={\frac{\left|\begin{array}{cccc} -T_{15} & T_{12} & T_{13} & T_{14}\\ -T_{25} & T_{22} & T_{23} & T_{24}\\ -T_{35} & T_{32} & T_{33} & T_{34}\\ -T_{45} & T_{42} & T_{43} & T_{44} \end{array}\right|}{\left|\begin{array}{c} T_{11}\;\;T_{12}\;\;T_{13}\;\;T_{14}\\ T_{21}\;\;T_{22}\;\;T_{23}\;\;T_{24}\\ T_{31}\;\;T_{32}\;\;T_{33}\;\;T_{34}\\ T_{41}\;\;T_{42}\;\;T_{43}\;\;T_{44} \end{array}\right|}|}_{\lambda =\pm \lambda_{i}}\beta_{i}={\frac{\left|\begin{array}{cccc} T_{11} & -T_{15} & T_{13} & T_{14}\\ T_{21} & -T_{25} & T_{23} & T_{24}\\ T_{31} & -T_{35} & T_{33} & T_{34}\\ T_{41} & -T_{45} & T_{43} & T_{44} \end{array}\right|}{\left|\begin{array}{c} T_{11}\;\;T_{12}\;\;T_{13}\;\;T_{14}\\ T_{21}\;\;T_{22}\;\;T_{23}\;\;T_{24}\\ T_{31}\;\;T_{32}\;\;T_{33}\;\;T_{34}\\ T_{41}\;\;T_{42}\;\;T_{43}\;\;T_{44} \end{array}\right|}|}_{\lambda =\pm \lambda_{i}}\;\gamma_{i}={\frac{\left|\begin{array}{cccc} T_{11} & T_{12} & -T_{15} & T_{14}\\ T_{21} & T_{22} & -T_{15} & T_{24}\\ T_{31} & T_{32} & -T_{15} & T_{34}\\ T_{41} & T_{42} & -T_{15} & T_{44} \end{array}\right|}{\left|\begin{array}{cccc} T_{11} & T_{12} & T_{13} & T_{14}\\ T_{21} & T_{22} & T_{23} & T_{24}\\ T_{31} & T_{32} & T_{33} & T_{34}\\ T_{41} & T_{42} & T_{43} & T_{44} \end{array}\right|}|}_{\lambda =\pm \lambda_{i}}\xi_{i}={\frac{\left|\begin{array}{cccc} T_{11} & T_{12} & T_{13} & -T_{15}\\ T_{21} & T_{22} & T_{23} & -T_{25}\\ T_{31} & T_{32} & T_{33} & -T_{35}\\ T_{41} & T_{42} & T_{43} & -T_{45} \end{array}\right|}{\left|\begin{array}{cccc} T_{11} & T_{12} & T_{13} & T_{14}\\ T_{21} & T_{22} & T_{23} & T_{24}\\ T_{31} & T_{32} & T_{33} & T_{34}\\ T_{41} & T_{42} & T_{43} & T_{44} \end{array}\right|}|}_{\lambda =\pm \lambda_{i}}$$


On the basis of Equation (60) and the superposition principle, *a_i_* and *d_i_* (*i* = 1–6) are used to represent arriving and departing wave vectors, and the displacement wave solutions of the laminated beam based on Q3DBT are expressed as
(62)
$${\bar{u}}_{0}(x)=\sum_{i=1}^{6}\left(\alpha_{ai}a_{i}e^{\lambda_{i}x}+\alpha_{di}d_{i}e^{-\lambda_{i}x}\right)$$

(63)
$${\bar{w}}_{0}(x)=\sum_{i=1}^{6}\left(\beta_{ai}a_{i}e^{\lambda_{i}x}+\beta_{di}d_{i}e^{-\lambda_{i}x}\right)$$

(64)
$${\bar{w}}_{0}(x)=\sum_{i=1}^{6}\left(\beta_{ai}a_{i}e^{\lambda_{i}x}+\beta_{di}d_{i}e^{-\lambda_{i}x}\right)$$

(65)
$${\bar{w}}_{z}(x)=\sum_{i=1}^{6}\left(\xi_{ai}a_{i}e^{\lambda_{i}x}+\xi_{di}d_{i}e^{-\lambda_{i}x}\right)$$

(66)
$$\bar{\phi }(x)=\sum_{i=1}^{6}\left(a_{i}e^{\lambda_{i}x}+d_{i}e^{-\lambda_{i}x}\right)$$


### 3.2. Phase and Scattering Relationship

As shown in Figure 2, dual coordinate systems are assumed to be established at nodes 1 and 2, which are located at either end of the beam. The fundamental principle of the MRRM is to establish a set of dual coordinate systems on the linear structure. By formulating the relationship between generalized displacements and generalized forces within each coordinate system, the phase matrix and scattering matrix of the structure are derived. Consequently, a reverberation-ray matrix that incorporates the dynamic characteristics of the structure is obtained for subsequent analysis. The coordinate transformation can be regarded as being symmetric about the *x* axis. The *x*^12^*z*^12^ coordinate system at node 1 is defined as being parallel to the positive direction. Clearly, the stiffness coefficient of the material and the arrangement of the layers will be reversed under the *x*^12^*z*^12^ coordinate system. In the following derivation, the superscripts 12 and 21 are used to represent physical quantities in a single coordinate system, respectively. Thus, every coefficient in Equations (27)–(32) and (43) under x^21^z^21^ can be obtained as

(67)
$$A_{11}^{21}=A_{11}^{12},\;B_{11}^{21}=-B_{11}^{12},\;D_{11}^{21}=D_{11}^{12},\;M_{11}^{21}=-M_{11}^{12},\;N_{11}^{21}=N_{11}^{12},\;O_{11}^{21}=O_{11}^{12}$$


(68)
$$J_{11}^{21}=-J_{11}^{12},\;K_{11}^{21}=K_{11}^{12},\;L_{11}^{21}=L_{11}^{12},\;G_{33}^{21}=G_{33}^{12},\;O_{55}^{S21}=O_{55}^{S12}$$


(69)
$$I_{0}^{21}=I_{0}^{12},\;I_{1}^{21}=-I_{1}^{12},\;I_{2}^{21}=I_{2}^{12},\;I_{3}^{21}=-I_{3}^{12},\;I_{4}^{21}=I_{4}^{12},\;I_{5}^{21}=I_{5}^{12},\;I_{6}^{21}=I_{6}^{12},\;I_{7}^{21}=I_{7}^{12}$$


On the basis of the specified positive direction, the generalized displacements and force vectors under *x*^21^*z*^21^ can clearly be written as
(70)
$$\delta^{21}={\left\{-{\bar{u}}_{0},-{\bar{w}}_{0},{\bar{\theta }}_{xb},{\bar{\theta }}_{xs},-{\bar{w}}_{z},\bar{\phi }\right\}}^{T}$$

(71)
$$f^{21}={\left\{{\bar{N}}_{xx},{\bar{Q}}_{xxb},-{\bar{M}}_{xxb},-{\bar{M}}_{xxs},{\bar{Q}}_{xxz},\int_{-h/2}^{h/2}{\bar{D}}_{x}\cos(\alpha z)dz\right\}}^{T}$$


Meanwhile, the physical properties of a certain point *x*^12^ on the beam are consistent. This is true even in the dual coordinate system. By combining Equations (60), (61), (70), and (71), we can obtain
(72)
$$\delta^{12}(x^{12})=T_{\delta }\delta^{21}(L-x^{12})$$

(73)
$$f^{12}(x^{12})=T_{\delta }f^{21}(L-x^{12})$$

where substituting Equations (72) and (73) into Equations (60) and (61) yields
(74)
$$A_{\delta }^{21}P(0)a^{21}+D_{\delta }^{21}P(0)d^{21}=T_{\delta }\left[A_{\delta }^{12}P(-L)a^{12}+D_{\delta }^{12}P(L)d^{12}+q_{\delta }^{12}(L)\right]$$

(75)
$$A_{f}^{21}P(0)a^{21}+D_{f}^{21}P(0)d^{21}=T_{f}\left[A_{f}^{12}P(-L)a^{12}+D_{f}^{12}P(L)d^{12}+q_{f}^{12}(L)\right]$$


By classifying the vectors {*a*^12^, *a*^21^} and {*d*^12^, *d*^21^}, Equations (74) and (75) can be rewritten as
(76)
$$\begin{array}{ll} \left[\begin{array}{l} a^{12}\\ a^{21} \end{array}\right]\; & =\underset{P_{h}}{\underbrace{{\left[\begin{array}{cc} -T_{\delta }A_{\delta }^{12}P(-L) & A_{\delta }^{21}P(0)\\ -T_{f}A_{f}^{12}P(-L) & A_{f}^{21}P(0) \end{array}\right]}^{-1}\left[\begin{array}{cc} T_{\delta }D_{\delta }^{12}P(L) & -D_{\delta }^{21}P(0)\\ T_{f}D_{f}^{12}P(L) & -D_{f}^{21}P(0) \end{array}\right]}}\left[\begin{array}{l} d^{12}\\ d^{21} \end{array}\right]\\ & +\underset{q_{p}}{\underbrace{{\left[\begin{array}{cc} -T_{\delta }A_{\delta }^{12}P(-L) & A_{\delta }^{21}P(0)\\ -T_{f}A_{f}^{12}P(-L) & A_{f}^{21}P(0) \end{array}\right]}^{-1}\left[\begin{array}{c} T_{\delta }q_{\delta }^{12}(L)\\ T_{f}q_{f}^{12}(L) \end{array}\right]}} \end{array}$$

where **P***_h_* is a matrix containing the phase relationship of the structure. Ensuring numerical stability generally involves selecting the positive wave number among the wave numbers (±*λ_i_*) that appear in pairs.

In order to analyze dynamic problems, a set of coupling balance relationships needs to be supplemented at each node, in addition to the phase relationship of the structure. The transformation relationship between the local and global coordinate systems is as follows:
(77)
$$T_{G\delta }^{12}=T_{Gf}^{12}=diag\left\{1,1,1,1,1,1\right\}$$

(78)
$$T_{G\delta }^{21}=T_{Gf}^{21}=diag\left\{-1,-1,1,1,-1,1\right\}$$

where the subscript “*G*” represents the global coordinate system. Combining the generalized displacement and force vectors at node 1 through Equations (60), (61), (77), and (78) yields
(79)
$$T_{G\delta }^{12}\left(A_{\delta }^{12}P(0)a^{12}+D_{\delta }^{12}P(0)d^{12}\right)=u_{1}$$

(80)
$$T_{G\delta }^{21}\left(A_{\delta }^{21}P(0)a^{21}+D_{\delta }^{21}P(0)d^{21}\right)=u_{2}$$

(81)
$$T_{Gf}^{12}\left(A_{f}^{12}P(0)a^{12}+D_{f}^{12}P(0)d^{12}\right)+q_{1}=K_{1}u_{1}$$

(82)
$$T_{Gf}^{21}\left(A_{f}^{21}P(0)a^{21}+D_{f}^{21}P(0)d^{21}\right)+q_{2}=K_{2}u_{2}$$

where **u** and **q** represent the nodal state vectors. Equations (79) and (80) are substituted into Equations (81) and (82) to eliminate ***u***_1_ and ***u***_2_ as
(83)
$$K_{1}\left(A_{\delta }^{12}a^{12}+D_{\delta }^{12}d^{12}\right)=\left(A_{f}^{12}a^{12}+D_{f}^{12}d^{12}\right)+{\left(T_{G\delta }^{12}\right)}^{-1}q_{1}$$

(84)
$$K_{2}\left(A_{\delta }^{21}a^{21}+D_{\delta }^{21}d^{21}\right)=\left(A_{f}^{21}a^{21}+D_{f}^{21}d^{21}\right)+{\left(T_{G\delta }^{21}\right)}^{-1}q_{2}$$


Considering mechanical boundary conditions, ***K***_1_ and ***K***_2_ are the corresponding spring stiffness matrixes related to the two ends of the laminated beam. The stiffness matrixes of various support conditions can be found in Table 1. After classifying the {***a***^12^, ***a***^21^} and {***d***^12^, ***d***^21^} in Equations (83) and (84), the scattering relation is simplified as
(85)
$$\begin{array}{l} \left[\begin{array}{l} d^{12}\\ d^{21} \end{array}\right]=\underset{S}{\underset{⎵}{\left[\begin{array}{cc} {\left(K_{E}^{1}D_{\delta }^{12}-D_{f}^{12}\right)}^{-1}\left(A_{f}^{12}-K_{E}^{1}A_{\delta }^{12}\right) & 0\\ 0 & {\left(K_{E}^{2}D_{\delta }^{21}-D_{f}^{21}\right)}^{-1}\left(A_{f}^{21}-K_{E}^{2}A_{\delta }^{21}\right) \end{array}\right]}}\left[\begin{array}{l} a^{12}\\ a^{21} \end{array}\right]\\ +\underset{q_{s}}{\underset{⎵}{\left[\begin{array}{c} {\left(K_{E}^{1}D_{\delta }^{12}-D_{f}^{12}\right)}^{-1}{\left[T_{G}^{12}\right]}^{-1}q^{1}\\ {\left(K_{E}^{1}D_{\delta }^{21}-D_{f}^{21}\right)}^{-1}{\left[T_{G}^{21}\right]}^{-1}q^{2} \end{array}\right]}} \end{array}$$

where ***S*** and ***q***_s_ represent the global scattering matrix and nodal external excitation wave vector, respectively.

### 3.3. Natural Frequency Analysis

Obviously, the phase and scattering relationship contain the same elements ***a*** and ***d***. The overall MRRM equation can be derived by substituting Equation (76) into Equation (85) as
(86)
$$\left(E-R\right)d=s$$

where **R** = **SP***_h_* and **s** = **Sq***_p_* + **q***_s_*, respectively, contain the various dynamic characteristics of the system and the wave source vector generated by external excitation. For natural frequency analysis, ***s*** can be set to zero because external excitation is ignored. Therefore, the elements in matrix ***R*** are all complex functions related to frequency *ω* as
(87)
$$\det\left[E-R(\omega )\right]=0$$


The solution of the nonlinear equation is complicated and cannot use general mathematical solution methods. However, the analysis of Equation (87) indicates that zero is always the minimum value of the determinant when a certain *ω* is found. Therefore, a calculation method based on the golden section search (GSS) algorithm is suitable for solving the natural frequency of the structure and Equation (88) is rewritten as
(88)
$$G(\omega )=\left|\det\left[E-R(\omega )\right]\right|$$


First, we define a frequency search range *ω*_0_–*ω*_max_, search step ∆*ω*, and two search frequencies *ω*_1_ = *ω*_0_ + ∆ω and *ω*_2_ = *ω*_1_ + ∆*ω*, and calculate the function values *G*(*ω*_1_) and *G*(*ω*_2_) to determine a frequency value that satisfies *G*(*ω*_1_) < *G*(*ω*_2_) and *G*(*ω*_0_) > *G*(*ω*_1_). Then, the GSS algorithm is used to search for *ω*_3_ again and calculate the function value until the max frequency ω_max_ is reached.

### 3.4. Transient Vibration Analysis

In the MRRM, the transient response of the structure is obtained by the inverse Laplace transform of the steady-state wave. From the local coordinate system *x^ij^*, the transient response at any point *x* can be written as
(89)
$$\begin{array}{ll} \delta (x^{ij})\; & =\frac{1}{2\pi }\int_{-\infty }^{+\infty }\delta (x^{ij})e^{st}ds\\ & =\frac{1}{2\pi }\int_{-\infty }^{+\infty }\left[A_{\delta }^{ij}P(-x^{ij})E_{a}^{ij}a+D_{\delta }^{ij}P(x^{ij})E_{d}^{ij}d+q_{\delta }^{ij}(x^{ij})\right]ds \end{array}$$

(90)
$$\begin{array}{ll} f(x^{ij})\; & =\frac{1}{2\pi }\int_{-\infty }^{+\infty }f(x^{ij})e^{st}ds\\ & =\frac{1}{2\pi }\int_{-\infty }^{+\infty }\left[A_{f}^{ij}P(-x^{ij})E_{a}^{ij}a+D_{f}^{ij}P(x^{ij})E_{d}^{ij}d+q_{f}^{ij}(x^{ij})\right]ds \end{array}$$


On the basis of Equation (85), the departing wave ***d*** is obtained as
(91)
$$d={\left(E-R\right)}^{-1}s$$


Obviously, singularity arises in the abovementioned inverse transformation. The Neumann series expansion method, as an alternative, is used to replace (**E** − **R**) ^−1^ as
(92)
$${\left(E-R\right)}^{-1}=E+R+R^{2}+\ldots +R^{N}$$


The highest power *N* corresponds to the number of scattering waves, including low- and high-frequency waves, observed during the time period *t*. Different propagation speeds make it difficult to select an appropriate value of *N*, but the effect of dispersive waves on wave speed is negligible over shorter time periods. The minimum propagation time of the travelling waves is recorded as *t*_min_. *N* should be an integer greater than *t*/*t*_min_, allowing travelling waves propagating in all paths to be recorded. After replacing (***E*** − ***R***) ^−1^ with the Newman series, the transient response at any point on the beam can be calculated.
(93)
$$\begin{array}{ll} \delta (x^{ij})\; & =\frac{1}{2\pi }\int_{-\infty }^{+\infty }\delta (x^{ij})e^{st}ds\\ & =\frac{1}{2\pi }\int_{-\infty }^{+\infty }\left\{\left[A_{\delta }^{ij}P(-x^{ij})E_{a}^{ij}P_{h}+D_{\delta }^{ij}P(x^{ij})E_{d}^{ij}\right]\sum_{0}^{N-1}R^{n}s+A_{\delta }^{ij}P(-x^{ij})E_{a}^{ij}q_{p}+q_{\delta }^{ij}(x^{ij})\right\}e^{st}ds \end{array}$$

(94)
$$\begin{array}{ll} f(x^{ij})\; & =\frac{1}{2\pi }\int_{-\infty }^{+\infty }f(x^{ij})e^{st}ds\\ & =\frac{1}{2\pi }\int_{-\infty }^{+\infty }\left\{\left[A_{f}^{ij}P(-x^{ij})E_{a}^{ij}P_{h}+D_{f}^{ij}P(x^{ij})E_{d}^{ij}\right]\sum_{0}^{N-1}R^{n}s+A_{f}^{ij}P(-x^{ij})E_{a}^{ij}q_{p}+q_{f}^{ij}(x^{ij})\right\}e^{st}ds \end{array}$$


## 4. Results Verification and Discussion

### 4.1. Result Verification

In this section, we will mainly verify the various dynamics results of the sandwich piezoelectric laminated beam. The comparison data are mainly from Ansys-V18.2, considering that few relevant reference data are available. The model is structured as two PZT layers with a base layer in between, and it ignores the bonding effect of the paste used to connect these layers. For the FEM analysis, the piezoelectric layer was modeled using SOLID5 elements, while the substrate layer was meshed with SOLID45 elements. The mesh was generated with a uniform element size of 0.01 m × 0.01 m × 0.01 m. Consequently, for the example presented in Table 2, the model comprises a total of 200 SOLID5 elements and 100 SOLID45 elements. The relevant material parameters are given in Table 1. In addition, all frequency parameters obtained by calculation and simulation are expressed in dimensionless form as 
$\Omega =\omega L\sqrt{{\left(\rho /c_{55}\right)}_{PZT}}$
.

In the first verification example, the total thickness of the beam with the laminated scheme of PZT/Base layer/PZT is set to 0.03 m. The analysis is conducted under two boundary conditions—clamped–clamped (C-C) and simply supported (S-S)—and across different slenderness ratios. In the first verification example, the total thickness of the beam with the laminated scheme of PZT/Base layer/PZT is set to 0.03 m under the two boundary conditions of C-C and S-S and different slenderness ratios. The external voltage and temperature are both set to zero. The frequency comparison results of the first six steps are given in Table 3. The addition of the Poisson effect in the *z* direction and the PZT material is generally a transversely isotropic material with a larger value of *c*_13_ and *c*_33_. Thus, a certain error with the finite element is generated when the slenderness ratio is less than 30 in this sandwich laminated beam model. However, this error can be controlled within 2%. The Poisson effect becomes insignificant when the slenderness ratio is greater than 30. All obtained results are in good agreement with the finite element, and the error is controlled within 1%.

Next, the effect of different temperature rises on frequency parameters under the abovementioned three-laminated structure will be verified. Meanwhile, C-C and C-S are selected as the boundary, given that the thermal stress is mainly generated in the axial direction. The length is set as 1 m, and a 10 kV external voltage is applied to the upper and lower poles. In Ansys 18.2, the model comprises a beam with a length of 1 m and a width of 0.01 m. The thickness of the piezoelectric ceramic layer and the width of the substrate layer are both 0.01 m. The three layers are bonded together through the paste command. Table 2 counts the frequency parameters of the temperature rise from 25 °C to 100 °C, and the results are also very close to those of FEM. The frequency parameters decrease as the temperature rises, considering that a higher temperature will reduce the stiffness of the material. Furthermore, the effect of externally applied voltage on the free vibration is very slight.

According to the overall Equation (85), MRRM is also suitable for analyzing the transient response according to the wave source vector s. The time t is set as 15 ms, the incremental step is 1000, a unit step loading applied in the *z* direction acts at *L*/4, and the observation point is set at 3*L*/4 to show the excitation response more clearly and accurately. Herein, the length of the beam is specified as 1 m, with the overall thicknesses set to 0.04 m and 0.05 m, respectively, and the thickness of each layer is assumed to be equal. Clamped–clamped (C-C) boundary conditions are uniformly adopted to ensure the activation of thermal stress, and the applied thermal environments are uniform thermal fields at 25 °C and 100 °C. In the finite element simulation, the pre-stress inside the structure under different temperature rises is first calculated and stored in the previous analysis step, and then the subsequent transient thermal vibration response is derived based on this pre-stress. The response curves and FEM results under different slenderness ratios are shown in Figure 3a. The thermal transient responses under ∆*T* = 25 °C and ∆*T* = 100 °C are also compared in Figure 3b, and they are in good agreement with FEM results.

The above-mentioned three comparative examples indicate that MRRM is a highly accurate and efficient analysis method. Furthermore, very convenient adjustment interfaces are provided for the material properties, geometric properties, and external environmental parameters of the analyzed structure to greatly improve the efficiency of parameterized research.

### 4.2. Case 1: Base Layer Properties Under DIFFERENT Elastic Boundaries in Thermal Environment

Sandwich piezoelectric laminated structures are commonly used in engineering fields such as micro ultrasonic motors and energy aids. Compared with an ordinary composite laminated structure, they have better design properties, such as different material properties, geometric properties, and lamination method of base layers, which can obviously affect the dynamic performance. In this section, we mainly analyze the free and transient vibration characteristics of the sandwich piezoelectric laminated beam from the aspects including spring stiffness coefficient, elastic modulus, slenderness ratio, thickness ratio, fiber orientation, and external temperature rise.

The advantage of using springs to simulate boundary forces is that they can break away from the limitations of classical boundary conditions and obtain results closer to those in actual engineering. The effect of the change of the stiffness coefficient of each spring on the frequency parameters is studied separately to better express the numerical results. The stiffness matrices are {*k_u_*_0_, 10^18^, 10^18^, 10^18^, 10^18^}, {10^18^, *k_w_*_0_, 10^18^, 10^18^, 10^18^}, {10^18^, 10^18^, *k_xb_*, 10^18^, 10^18^}, {10^18^, 10^18^, 10^18^, *k_xs_*, 10^18^}, and {10^18^, 10^18^, 10^18^, 10^18^, *k_wz_*}. The stiffness coefficients represented by {*k_u_*_0_, *k_w_*_0_, *k_xb_*, *k_xs_*, *k_wz_*} increase from 0 to 10^18^. Since the frequency of each order corresponding to *k*_*u*0_ does not change, the changing law of {*k_w_*_0_, *k_xb_*, *k_xs_*, *k_wz_*} is shown in Figure 4a–d. Evidently, Figure 4a–d illustrate that the influence of boundary spring stiffness on each order of natural frequency is concentrated within the range of 10^4^–10^8^. Since *k_xb_* is expressed as a function of *w*_0_, and *k*_xs_ is defined as the rotational component of the beam, both exhibit a similar sensitivity range with respect to the out-of-plane vibration of the beam. It is noteworthy that the spring stiffness corresponding to the thickness stretching degree of freedom *w_z_*, which is associated with the Poisson effect, does not exhibit frequency sensitivity and exerts no influence on the system frequency. When the spring stiffness is below or above this range, the structure degenerates into free and fixed boundary conditions, respectively. In addition, Figure 4e–g compare the first three orders of natural frequency values induced by variations in each spring stiffness, which clearly demonstrates the variation range and sensitivity of the system frequency when the spring stiffnesses corresponding to *w*_0_,
$\theta_{xb}$
, and 
$\theta_{xs}$
 change. Obviously, the main change interval of the frequency is concentrated between 10^5^–10^10,^ and the corresponding spring stiffness coefficient gradually increases with the rise in the order. Figure 4e–g also show that the change in *k_w_* has a more obvious influence on the frequency than *k_xb_* and *k_xs_*, while *k_wz_* is nearly very weak.

On the basis of this conclusion, the dynamic performance under any combination of boundary conditions can be studied. In the first example, we first discuss the change trend of the corresponding frequency parameters under different combination boundaries. Then, we add the elastic modulus of the base layer and the influence of different temperature rises. According to the interval range in Figure 4, E1 is specified as the *z* direction subject to elastic constraints, and E2 is the rotation subject to elastic constraints. The corresponding stiffness matrices are given in Table 4. An elastic modulus range from 10 GPa to 350 GPa is given for the base middle layer, and the Poisson’s ratio and density are kept constant. Figure 5 counts the frequency parameters under the four boundaries of C-C, E1-E1, E2-E2, and E1-E2. Evidently, different elastic boundaries significantly affect the free vibration. Compared with the classical boundaries C, S, and F in previous studies, the stiffness of the spring can also be a major design indicator, and the dynamic parameters under an arbitrary boundary can be studied more widely.

Various temperature rises from 0 °C to 200 °C are applied to the abovementioned laminated beams under E1-E1, E2-E2, and E1-E2 to study the effect of external temperature rise. Figure 6 shows the change curves of the first three-order frequencies. The frequency will show a significant downward trend as the temperature rises. An obvious buckling occurs when the temperature rise is greater than 100 °C because the Poisson effect is considered, and the coefficient *c*_13_ of PZT is generally larger.

### 4.3. Case 2: Geometric Parameters and Layering Methods of Laminated Beams

Case 1 implies that the material parameters of the middle base layer have less influence on the free vibration, and the trend of increase or decrease also tends to be linear. Therefore, the geometric characteristics of sandwich piezoelectric laminated beams should also be introduced. More accurate results will be obtained, especially for medium-thick beams with *L*/*h* ≤ 20, given that Q3D shear theory considers the effect of thickness stretching. A constant thickness of the beam is set as 0.01 m, and the elastic modulus of the base layer is 242 GPa. The numerical trend with slenderness ratio from 10 to 100 under the four boundaries is given in Figure 7a. Except for the first-order frequency under E1-E1, each group of frequency parameters decreases with the increase in the slenderness ratio, especially between 10 and 40, and the downward trend tends to be flat after *L*/*h* is greater than 40. The numerical difference under different boundary conditions also decreases as the slenderness ratio increases. When it is greater than 80 for the slender beam, the difference between the results of each group is very small, which means that slender beams are less affected by boundary conditions, but thick beams are more affected.

We will also study the frequency parameters with various thickness ratios *h_m_*/*h_c_* under different slenderness ratios. Several thickness ratios from 0.5 to 50 are selected on the *x* axis in Figure 7b. The value of the thickness ratio will not be selected very small because the piezoelectric sheet is generally very thin in engineering design. In the selected interval, the frequency parameters increase as the thickness ratio increases. The effect is more evident between 0.5 and 10 and becomes relatively stable after it is greater than 10. Meanwhile, a medium-thick beam with a smaller slenderness ratio has a higher frequency.

In previous research cases, the middle base layer of the sandwich piezoelectric laminate structure generally uses a single-layer isotropic material. The vibration characteristics of the composite material as the base layer are also rarely investigated. Therefore, different lamination methods of composite base layers composed of orthotropic materials should be introduced for parameter research. Figure 7c shows the frequency curve of the six lamination methods in different fiber orientations. Symmetrical and asymmetrical lamination methods including *θ*, 0/*θ*, *θ*/0/*θ*, 0/*θ*/0/*θ*, 0/*θ*/*θ*/0, and *θ*/0/*θ*/0/*θ* are considered. *θ* represents the fiber orientation from 0° to 180° with the x axis. The ratio of the total thickness of all composite layers to each piezoelectric sheet is set to 8, and C-C is the boundary condition. The frequency curve obtained is symmetrical about x = 90°. The range of variation is larger between 0° and 60°, and then it will slow down when it is close to 90°. The frequency parameters of [0/*θ*/*θ*/0] are higher than those of other lamination methods, while the single layer is the lowest.

### 4.4. Case 3: Parametric Analysis of Transient Vibration

This section discusses various transient response characteristics of sandwich piezoelectric beams. In the following investigations, a unit lateral load and the observation point are located at 1/4 and 3/4 of the beam, respectively, and the time increment is 0.01 or 0.02 s. The response changes of five spring stiffnesses are obtained to determine the elastic boundaries E1 and E2 of this part. Figure 8 summarizes the response curves of the five spring stiffnesses from 0 to 10^15^. The main change interval of the transverse spring *k*_*w*0_ is between 10^6^ and 10^8^, the rotary spring *k_wb_* is between 10^3^ and 10^6^, and the *k_xs_* is between 10^5^ and 10^9^. In addition, changes in *k_u_* and *k_wz_* have no effect on the transient response. Therefore, E1 and E2 in this section can be set as

E1: *diag*{10^18^, 10^7^, 10^18^, 10^18^, 10^7^, 0}

E2: *diag*{10^18^, 10^18^, 10^5^, 10^6^, 10^18^, 0}

**Figure 8 materials-19-00136-f008:**
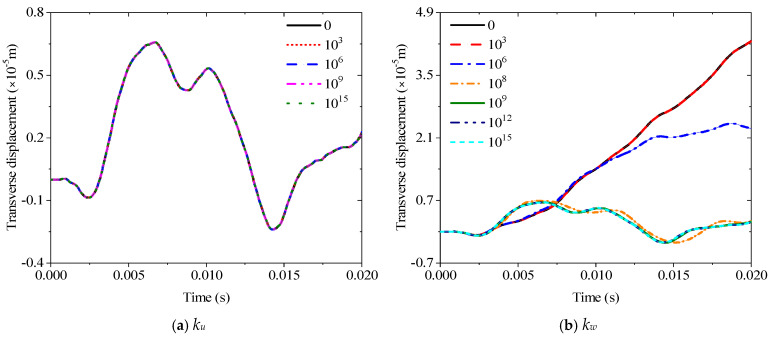
Variations of the transverse displacement responses versus the elastic restraint parameters for a sandwich laminated beam (PZT-5A/Al/PZT-5A): (**a**) *k_u_*; (**b**) *k_w_*; (**c**) *k_xb_*; (**d**) *k_xs_*; (**e**) *k_wz_*.

The first example mainly discusses the effect of different thickness ratios *h_m_*/*h_p_* from 1 to 10 under four kinds of boundary conditions: C-C, C-S, E1-E2, and E2-E2, and all results are obtained by MRRM programming. Figure 9 shows that the increase in the thickness ratio under all boundaries negatively affects the amplitude of the transient response. Notably, the amplitude and period of the response curve under the two elastic boundaries have become larger. The second example involves the transient response of the fiber orientation of an orthotropic base layer. The angle can only be considered from 0° to 90° due to the symmetry. The thickness ratio is set to 4 in accordance with the previous study. The calculation results in Figure 10 show that the fiber orientation positively affects the response amplitude, which means the rigidity of the laminated beam will be weakened when the angle increases. This effect is mainly reflected in the range of 0–45°, and the amplitude increase between 45° and 90° is significantly reduced. The elastic boundaries E1 and E2 will also increase the response period under this lamination method. Figure 11 shows the response curves of thermal stress under various boundaries. As shown in the figure, the effect of thermal stress in the first quarter of the calculation time is very small and nearly negligible. The amplitude of the response gradually increases with the rise in temperature with the extension of time. This trend in E2-E2 is the most obvious.

## 5. Conclusions

Using the Q3DBT and MRRM theories, this research examines the free vibration characteristics and transient dynamic response of sandwich piezoelectric laminated beams under elastic boundary constraints. Considering the Poisson and thermal effects, we use MRRM to uniformly list the derived thermoelectric coupling differential equations. The Poisson effect refers to the thickness stretching effect in structures, specifically the inherent property of materials to contract laterally when subjected to unidirectional stress, thereby releasing the constraint on deformation in the thickness direction. In Quasi-3D beam theory (Q3DBT), this is achieved by introducing an additional displacement field function along the thickness direction. The modal and transient values obtained by the GSS search algorithm and Newman series expansion are compared with FEM simulation to verify the accuracy and efficiency of this method for analyzing sandwich piezoelectric beams. Good consistency is achieved. Furthermore, several numerical examples are developed, including the stiffness coefficient of each support spring, the parameters of the base material, geometric properties, different laminated methods, and external temperature rise. The following conclusions are obtained from the results:(1)The support stiffness of the spring will significantly affect the natural frequency and transient response characteristics of the sandwich piezoelectric laminated beam within the range of 10^4^–10^8^ N/m. When the stiffness is lower or higher than this range, the structure will be subjected to traditional free or fixed constraint boundary conditions.(2)The Poisson effect of the sandwich piezoelectric laminated beam will be prominent when L/h < 30. Selecting an intermediate metal layer with a higher elastic modulus can effectively enhance the stiffness of the beam and improve its natural frequency. Meanwhile, metals with a higher elastic modulus will induce greater thermal stress, which will reduce the stiffness of the beam within the linear elastic range.(3)The length of the beam will reduce the sensitivity of the system frequency to boundary conditions, and beams with a slenderness ratio greater than 60 can maintain a relatively stable frequency range under most boundary conditions. In addition, the thickness ratio between the metal layer and the piezoelectric layer within 1–10 can significantly affect the natural frequency of the beam; a further increase in the thickness ratio will homogenize the sandwich beam, thereby reducing the range of its frequency variation.(4)The sandwich laminated beam with composite materials as the interlayer will achieve the minimum stiffness when the fiber orientation is 90°. Moreover, the dynamic response of the beam is highly sensitive to the layup modes in the range of 0–60°. Through the rational arrangement of fiber angles and layup modes, the output of the sandwich-type piezoelectric actuator can be effectively controlled.(5)The thermal environment will generate axial thermal stress in linear beam structures, which will weaken the structural stiffness and lead to an increase in the amplitude of dynamic responses. By reasonably increasing the constraint stiffness at both ends of the beam, the influence of thermal stress on transient vibration can be effectively reduced.(6)Due to current laboratory limitations, experimental validation under corresponding conditions was not feasible in this study. We recognize the importance of such validation and regard it as a key objective for our future work.(7)The FEM simulations employed solid elements, which introduced modeling assumptions related to width, even though only one mesh element was used in the y-direction. In contrast, the model in this study is a one-dimensional linear formulation, which does not account for modes in the y-direction at all.(8)Factors such as the form and magnitude of the loading, as well as the control of dynamic response via piezoelectric materials, represent limitations of the current work and warrant further investigation.

## Figures and Tables

**Figure 1 materials-19-00136-f001:**
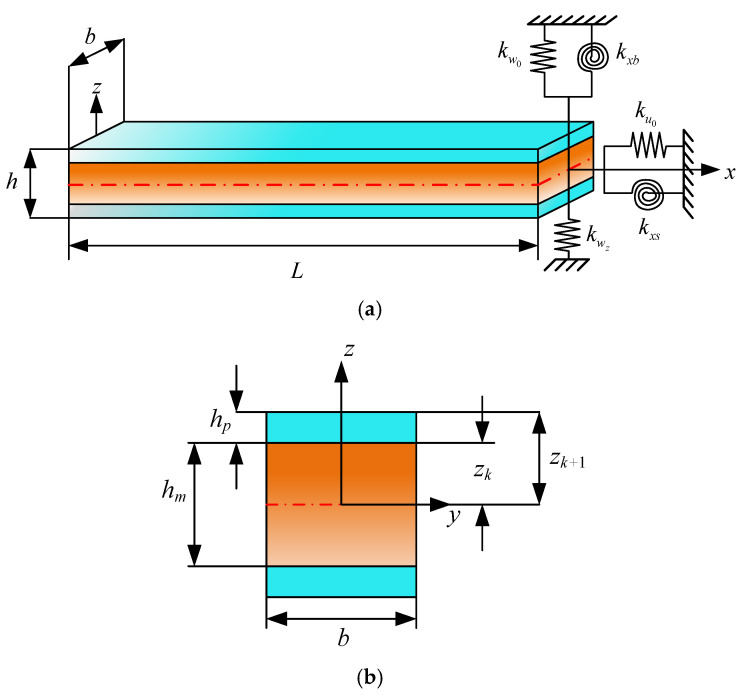
Schematic diagram of sandwich piezoelectric laminated beam: (**a**) The sandwich piezoelectric laminated beam with general boundary condition; (**b**) Thickness distributed for the section.

**Figure 2 materials-19-00136-f002:**
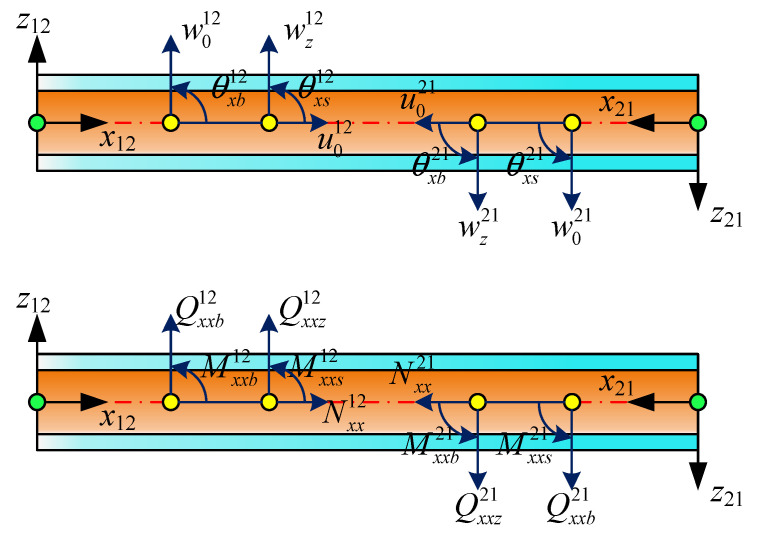
Dual coordinate system of the laminated beam.

**Figure 3 materials-19-00136-f003:**
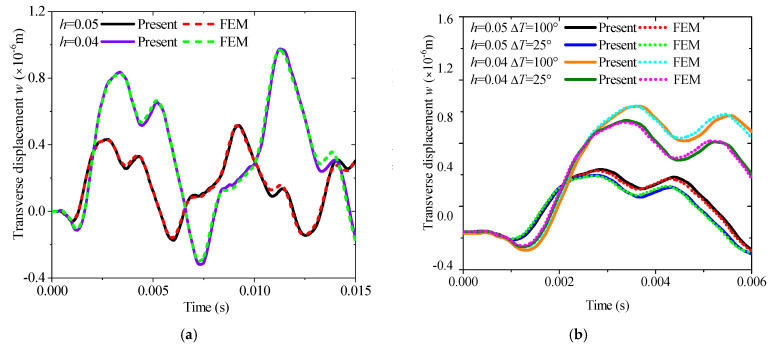
Comparison of the transient transverse displacement responses with different thickness *h* and temperature rises ∆*T* (*L* = 1 m, C-C): (**a**) different thickness ratios; (**b**) different temperature rise.

**Figure 4 materials-19-00136-f004:**
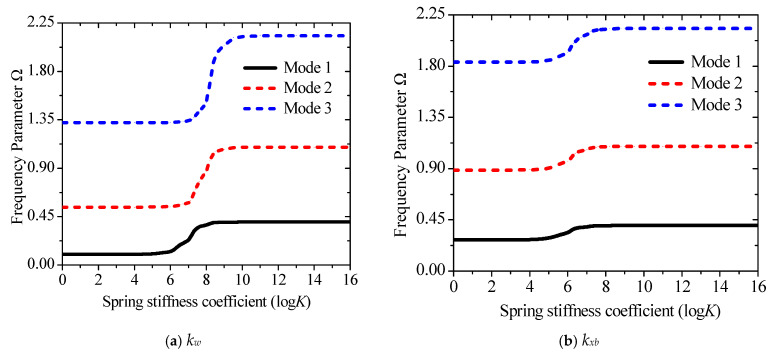
Variations of the frequency parameter Ω versus the elastic restraint parameters for a sandwich laminated beam (PZT-5A/Al/PZT-5A): (**a**) *k_w_*; (**b**) *k_w__b_*; (**c**) *k_xs_*; (**d**) *k_w__z_*; (**e**) Mode 1; (**f**) Mode 2; (**g**) Mode 3.

**Figure 5 materials-19-00136-f005:**
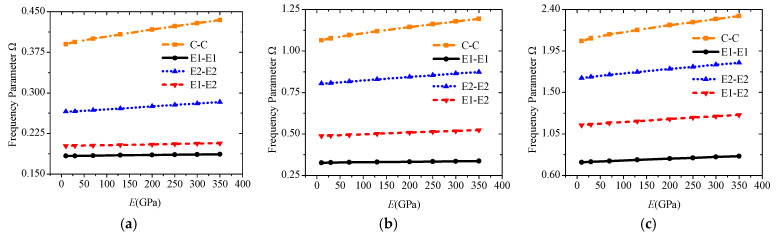
Variations of the frequency parameter Ω versus boundary conditions and elastic modulus for a sandwich laminated beam (PZT-5A/Material-A/PZT-5A, *L* = 1 m, *h* = 0.03 m): (**a**) Mode 1; (**b**) Mode 2; (**c**) Mode 3.

**Figure 6 materials-19-00136-f006:**
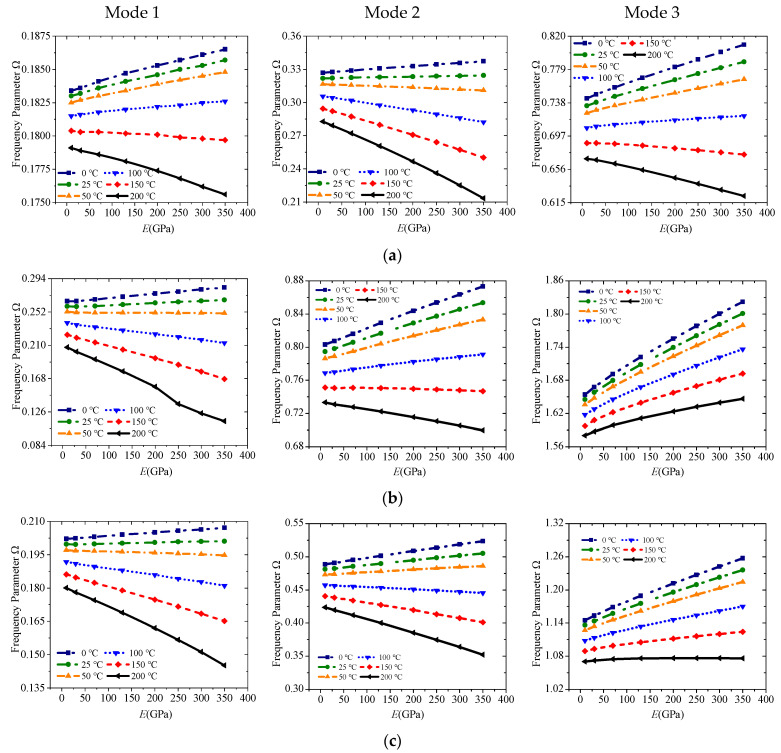
Variations of the frequency parameter Ω versus boundary conditions and elastic modulus under different temperature rise for a sandwich laminated beam (PZT-5A/Material-A/PZT-5A, *L* = 1 m, *h* = 0.03 m): (**a**) E1-E1; (**b**) E2-E2; (**c**) E1-E2.

**Figure 7 materials-19-00136-f007:**
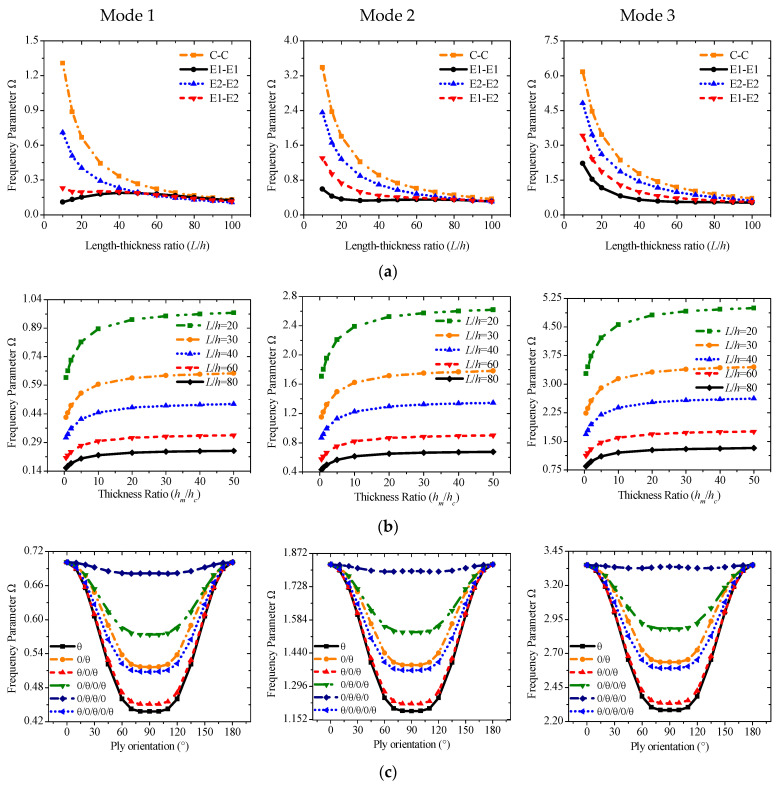
Variations of the frequency parameter Ω versus the length–thickness ratio (*L*/*h*), thickness ratio (*h_m_*/*h_p_*), and ply orientation (base layer) for a sandwich laminated beam. (*h* = 0.03 m, C-C): (**a**,**b**) PZT-5A/Al/PZT-5A; (**c**) PZT-5A/Material-B/PZT-5A.

**Figure 9 materials-19-00136-f009:**
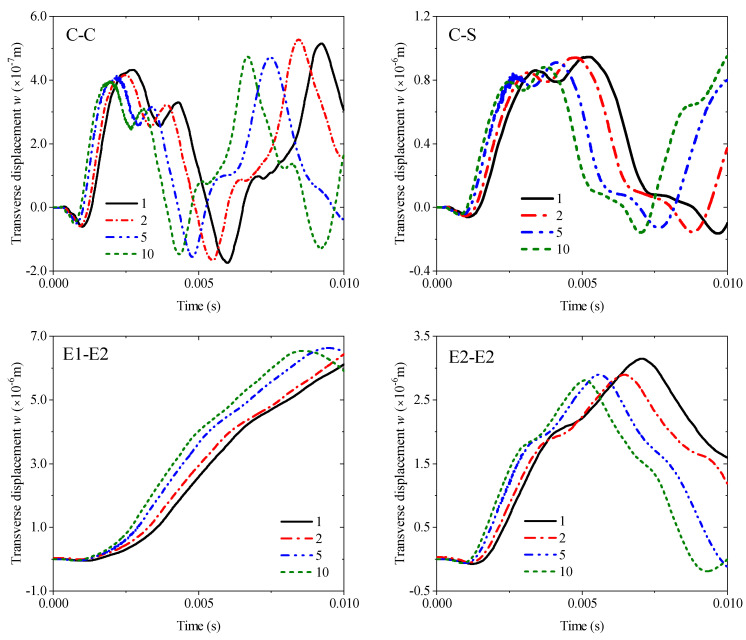
Variations of the transverse displacement responses versus the thickness ratio (*h_m_*/*h_p_*) and boundary conditions for a sandwich laminated beam (PZT-5A/Al/PZT-5A, *L* = 1 m, *h* = 0.05 m).

**Figure 10 materials-19-00136-f010:**
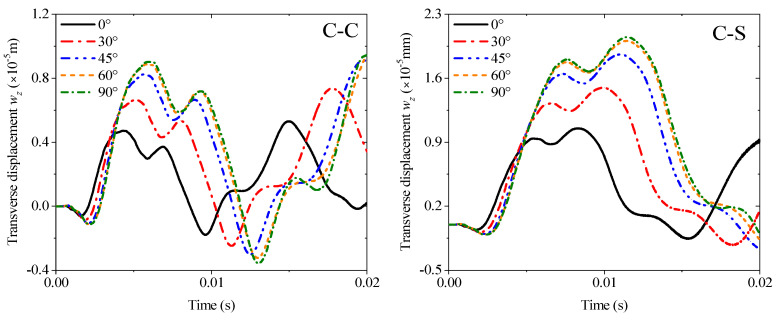
Variations of the transverse displacement responses versus the ply orientation (base layer) and boundary conditions for a sandwich laminated beam (PZT-5A/Material-B/PZT-5A, *L* = 1 m, *h* = 0.02 m).

**Figure 11 materials-19-00136-f011:**
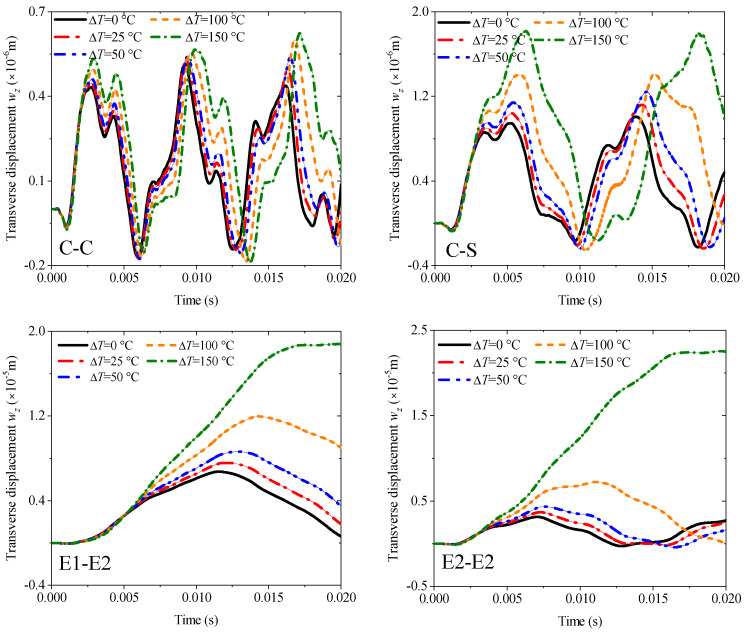
Variations of the transverse displacement responses versus the temperature rises ∆*T* and boundary conditions for a sandwich laminated beam (PZT-5A/Al/PZT-5A, *L* = 1 m, *h* = 0.05 m).

**Table 1 materials-19-00136-t001:** Related material parameters.

PZT-5A [57]
c_11_ = 121 GPa, c_13_ = 75.2 GPa, c_33_ = 111 GPa, c_55_ = 21.1 GPa, ρ = 7750 kg/m^3^, *α*_1_ = *α*_3_ = 2.68 × 10^−6^/K e_31_ = −5.4 C/m^2^, e_33_ = 15.8 C/m^2^, e_15_ = 12.3 C/m^2^ s_11_ = 8.107 nF/m, s_33_ = 7.34 nF/m
AI
c_11_ = 72.2 GPa, c_55_ = 26.94 GPa, *ρ* = 2700 kg/m^3^, *α*_1_ = *α*_3_ = 23.2 × 10^−6^/K
Material-A
*E* = Variable value, *μ* = 0.34, *ρ* = 2700 kg/m^3^, *α*_1_ = *α*_3_ = 6 × 10^−6^/K
Material-B
*E*_1_/*E*_2_ = 15, *E*_2_ = *E*_3_ = 10 GPa, *μ*_12_ = *μ*_13_ = *μ*_23_ = 0.3, G_12_ = G_13_ = 0.6 *E*_2_, G_23_ = 0.5 *E*_2_, *ρ* = 1600 kg/m^3^

**Table 2 materials-19-00136-t002:** Comparison of natural frequencies for C-C sandwich piezoelectric laminated beam under different temperature rises. (PZT-5A/Material-A/PZT-5A, *E* = 50 Gpa, *L* = 1 m, *h* = 0.03 m).

T		Mode					
		1	2	3	4	5	6
0	FEM	1.0797	2.0917	3.4077	5.0052	6.8605	8.9480
	Present	1.0868	2.1093	3.4428	5.0665	6.9568	9.0893
	Difference	0.66%	0.84%	1.03%	1.22%	1.4%	1.58%
25	FEM	1.0757	2.0872	3.4030	5.0005	6.8552	8.9427
	Present	1.0735	2.0944	3.4276	5.0506	6.9405	9.0726
	Difference	−0.2%	0.34%	0.72%	1.00%	1.24%	1.45%
50	FEM	1.0716	2.0829	3.3984	4.9957	6.8502	8.9378
	Present	1.0598	2.0799	3.4120	5.0346	6.9237	9.0558
	Difference	−1.11%	−0.14%	0.4%	0.78%	1.07%	1.32%
75	FEM	1.0694	2.0803	3.3957	4.9930	6.8475	8.9347
	Present	1.0461	2.0651	3.3964	5.0186	6.9073	9.0387
	Difference	−2.18%	−0.73%	0.02%	0.51%	0.87%	1.16%
100	FEM	1.0636	2.0740	3.3890	4.9858	6.8403	8.9271
	Present	1.0323	2.0498	3.3807	5.0022	6.8906	9.0215
	Difference	−2.94%	−1.16%	−0.24%	0.33%	0.73%	1.06%

**Table 3 materials-19-00136-t003:** Comparison of natural frequencies for a sandwich piezoelectric laminated beam under different boundary conditions and slenderness ratios. (PZT-5A/Al/PZT-5A, *h* = 0.03 m).

BC	Slenderness Ratio		Mode					
C-C			1	2	3	4	5	6
	*L*/*h* = 20	FEM	0.6587	1.7793	3.3996	5.4522	7.8754	10.6117
		Present	0.6662	1.8049	3.4565	5.5525	8.0278	10.8181
		Difference	1.14%	1.44%	1.67%	1.84%	1.94%	1.94%
	*L*/*h* = 30	FEM	0.4420	1.2071	2.3377	3.8071	5.5894	7.6573
		Present	0.4448	1.2166	2.3588	3.8458	5.6518	7.7484
		Difference	0.64%	0.79%	0.9%	1.02%	1.12%	1.19%
	*L*/*h* = 60	FEM	0.2216	0.6095	1.1912	1.9612	2.9157	4.0498
		Present	0.2221	0.6114	1.1947	1.9678	2.9267	4.0658
		Difference	0.19%	0.3%	0.29%	0.34%	0.38%	0.39%
	*L*/*h* = 100	FEM	0.133	0.3663	0.7174	1.1841	1.7658	2.4611
		Present	0.1331	0.3667	0.7182	1.1856	1.7682	2.4649
		Difference	0.08%	0.1%	0.11%	0.13%	0.14%	0.15%
S-S	*L*/*h* = 20	FEM	0.2920	1.1547	2.5512	4.4281	6.7252	9.3824
		Present	0.2918	1.1538	2.5498	4.4258	6.7231	9.3693
		Difference	−0.08%	−0.08%	−0.05%	−0.05%	−0.03%	−0.14%
	*L*/*h* = 30	FEM	0.1951	0.7763	1.7320	3.0440	4.6892	6.6419
		Present	0.1950	0.7755	1.7306	3.0421	4.6868	6.6384
		Difference	−0.05%	−0.1%	−0.08%	−0.06%	−0.05%	−0.05%
	*L*/*h* = 60	FEM	0.0977	0.3898	0.8760	1.5526	2.4167	3.4640
		Present	0.0985	0.3898	0.8752	1.5513	2.4148	3.4615
		Difference	0.84%	−0.1%	−0.1%	−0.09%	−0.08%	−0.07%
	*L*/*h* = 100	FEM	0.0586	0.2344	0.5269	0.9357	1.4601	2.0989
		Present	0.0586	0.2342	0.5263	0.9348	1.4587	2.0971
		Difference	−0.01%	−0.08%	−0.12%	−0.1%	−0.09%	−0.09%

**Table 4 materials-19-00136-t004:** Boundary conditions and stiffness matrixes.

Boundary Conditions	Essential Conditions	Stiffness Matrixes
F	*N_xx_* = *Q_xxb_* = *M_xxb_* = *M_xxs_* = *Q_xxz_* = 0	*diag* (0, 0, 0, 0, 0, 0)
S	*N_xx_* = *w*_0_ = *M_xxb_* = *M_xxs_* = *w_z_* = 0	*diag* (0, 10^18^, 0, 0, 10^18^, 0)
C	*u*_0_ = *w*_0_ = *θ_xb_* = *θ_xs_* = *w_z_* = 0	*diag* (10^18^, 10^18^, 10^18^, 10^18^, 10^18^, 0)
E	*u*_0_, *w*_0_, *θ_xb_*, *θ_xs_*, *w_z_* ≠ 0	*diag* (*k*_u_, *k_w_*, *k_xb_*, *k_xs_*, *k_wz_*, *k_Φ_*)
E1	*u*_0_ = *θ_xb_* = *θ_xs_* = 0, *w*_0_ = *w_z_* ≠ 0	*diag* (10^18^, 10^7^, 10^18^, 10^18^, 10^7^, 0)
E2	*u*_0_ = *w*_0_ = *w_z_* = 0, *θ_xb_* = *θ_xs_* ≠ 0	*diag* (10^18^, 10^18^, 10^6^, 10^6^, 10^18^, 0)

## Data Availability

The original contributions presented in this study are included in the article. Further inquiries can be directed to the corresponding author.

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
