# Peer review of "Free and Transient Vibration Analysis of Sandwich Piezoelectric Laminated Beam with General Boundary Conditions"

_materials, 2025, doi:10.3390/ma19010136_

Round 1
Reviewer 1 Report
Comments and Suggestions for Authors
This manuscript presents a solution to the problem of free and transient vibration of a sandwich piezoelectric laminated beam. General boundary conditions are formulated for the beam under consideration. The manuscript is theoretically advanced and undoubtedly addresses an important issue. Below are some critical remarks.
- The introduction, especially its beginning, is written in an incomprehensible manner. One gets the initial impression that it's some kind of slang used exclusively by those deeply versed in the subject matter. This applies, among others, to lines 37-42 and 51-53. I suggest a more understandable introduction, explaining basic concepts with reference to the topic being addressed.
- Line 53. "... a two-dimensional bezm model". Here a certain doubt arises - what is a beam? The wording used in the manuscript is debatable.
- Lines 135-136 "The length of the beam is L and the thickness of each layer is equal ..."
This means that the three layers shown in Figure 1, for example, have identical thicknesses, even though the drawing itself indicates they are different thicknesses. I don't see the implications here, and I don't think that's what the authors intended. - Figure 1a. I would appreciate it if you could supplement the manuscript with an explanation of the physical meaning of these supports. The drawing shows that some of the spring supports are duplicated. Why are there two spring vertical supports, and why are there two spring rotational supports if they involve the same rotation? This is a very interesting and important issue, and in my opinion, crucial to understanding the rest of the manuscript.
- Figure 1b. In my opinion "z" is a coordinate, not a length, so I suggest showing "z_k" and "z_k+1" with one arrow.
- Equation (5). Please explain in the manuscript what Dx, Dy, Dz, D_z^T are.
- 𝐷𝑧𝑇 = 𝑝̄3𝛥𝑇 - it should be Equation (7).
- From an editorial perspective, the manuscript is poorly written. Symbols are sometimes italicized, sometimes in normal font, sometimes in a different font. This makes reading difficult. For example, on line 170: "matrixes e, E, and s," equation (7) uses italics. There are many such inconsistencies.
- Line 171. What do the symbols c11, c13, c33, e31, e33, ... mean?
What is a base layer? You can only guess at the end of the manuscript. - Line 181: δUM, line 183: φ0, line 186: δUE, line 191: Ii, line 209: bi,, lines 227, 228 and others.
- Equation (13). Integration over the beam width is omitted without any comment. I suggest some clarification in the manuscript.
- Line 199. Not all readers are experts in MRRM, so I propose some explanation of what these "wave solutions" are all about.
- Line 207. "Obviously, the following 207 12-order equation ...". I admit that I don't like it when someone uses the phrase "obviously", especially if it's not that obvious.
- I'm not entirely sure why the dual coordinate system of the laminated beam was introduced. There's probably a reason for it, but it doesn't seem to be well explained in the manuscript.
- Table 1. What does E=open mean?
- Improve in line 270: ana;ysis
- Lines 289, 290, 291: The symbol x is used on each line; each time a different font is used.
- Line 312 "boundary conditions of C-C and S-S" - explain what do C-C and S-S mean.
- Table 3. The authors compare their results with those of the FEM model, reporting "Error." In my opinion, this isn't an "Error," but a "Difference." It seems the models used aren't identical, and each model is subject to certain errors. This is simply a comparison of two results, with neither being better or worse. They are simply slightly different. However, the question arises: what numerical model was used, as nothing is explained in the manuscript.
- Lines 362-367. There are some inconsistencies, but no units. There are also incorrectly written numbers, e.g., 1018 instead of 10^18.
- Line 478. It is not explained what the authors mean by the Poisson effect.
- Line 502. What does PATENTS mean?
Authors often use confusing terms, as if to "short-cut" them, aimed at a narrow circle of specialists. This is probably justified, but it made it very difficult for me to understand the entire text. Below are a few of my concerns.
- Lines 112-113 "The previous studies of sandwich laminated beams have a defect on the boundary condition that most focused on clamped, simple support, and free."
I disagree with the phrase "have a defect on the boundary condition." It's not a defect if different support conditions were previously assumed. - Lines 135-136 "The length of the beam is L and the thickness of each layer is equal if no special instructions are provided to facilitate the derivation process."
I do not understand this sentence. - Lines 306-307 "The structure of the model is superimposed by a base layer with two PZT layers and ignores the paste effect between layers." I had to spend a moment trying to understand what the authors meant by this sentence. Could you rephrase it to make it more understandable?
- Line 476 "This study investigates the free vibration and transient response of sandwich piezoelectric laminated beams under elastic boundaries on ...". I know what the authors mean, but in my opinion this is an inappropriate formulation.
Author Response
Please refer to the attachment for the response to your suggestion.

Reviewer 2 Report
Comments and Suggestions for Authors 1. The study is mainly based on the numerical and computational model, but the experimental analysis could not be performed under similar conditions. 2. The abstract should refer to the similarities and errors in both the models. Provide numerical values (quantitative). 3. The ratio of layers in the sandwich are determined arbitrarily. Why was the ratio not optimized? provide reference to literature for this. 4. The equations should have references to original sources. 5. What are the materials? They should be described in detail. Not just their properties (Table 1). Are the properties measured or taken from literature? Provide references. 6. Why are the frequency parameter data points in Figure 5 and 6, joined by linear graphs?What is the regression coefficient? Why nonlinear curves not plotted? Justify and give appropriate explanations in statistical terms. Provide regression equations and coefficient of determination. 7. Why Is it nonlinear in Fig. 7? Justify and give appropriate explanations in statistical terms. Provide regression equations and coefficient of determination. 8. Explain the nature of transverse displacement in Fig. 7 to 9, Why are these different behaviors visible? 9. The conclusion is also generic and lacks the numerical data and parametric optimizations of the models. Provide quantitative data. It must be rewritten. 10. What are the Patents?? 11. provide a limitations section explaining why experimental data could not be generated. What are the sources of error?Author Response
Please refer to the attachment for the response to your suggestion.

Reviewer 3 Report
Comments and Suggestions for Authors
The authors present a thermo-electro-mechanical dynamic analysis of a sandwich piezoelectric laminated beam using: a Quasi-3D Shear Deformation Beam Theory (Q3DBT) with thickness-stretch effects; the Hamilton’s principle to derive governing equations; the Method of Reverberation-Ray Matrix (MRRM) to compute wave propagation, natural frequencies, and transient responses; the general elastic (spring) boundary conditions allowing simulation of classical and arbitrary support configurations. Based on results and concluding remarks, the work provides a rich parametric study useful for actuator/stator design in aerospace piezoelectric systems, and it extends the MRRM—a method rarely applied to piezoelectric laminated beams—to unified dynamic analysis including free vibration and transient response. The following revisions, however, are suggested, in order to improve the work before its possible acceptance:
- Please, provide some error percentages or convergence test.
- Validation figures and tables are missing in the extract, and it is not possible to check accuracy or robustness.
- The transient analysis lacks benchmarking against analytical or FEM time integration.
- There isn’t a clear study of how many wave modes are needed for stable MRRM solutions.
- Grammar and notation errors throughout text. Please, check and correct them.
- Insufficient clarity in some mathematical derivations. Please, improve it.
- When the authors talk about the computational strategies to solve the governing differential equations of the model, please, consider further recent and advanced references, for affinity reasons, e.g. DOI 10.1088/0964-1726/22/3/035006; DOI: 10.1016/j.compstruct.2024.118801; also applied to more complicated shell geometries.
Author Response

(The authors gave the same response as above.)

Reviewer 4 Report
Comments and Suggestions for Authors
Please find in the attached file comments on the manuscript Materials-3981985

Author Response

(The authors gave the same response as above.)

Reviewer 5 Report
Comments and Suggestions for Authors
The manuscript “Free and Transient Vibration Analysis of Sandwich Piezoelectric Laminated Beam with General Boundary Conditions” by Xiaoshuai Zhang and et al. is a scientific work where the authors study free vibration and transient response for sandwich piezoelectric laminated beam with elastic boundaries in thermal environment.
The abstract constitutes a meaningful representation of the contents of the article.
The Introduction constitutes an accurate representation of the previous works.
The theoretical basics are well represented.
The figures and tables are generally of a good quality, are clear, and legible. There are no further comments regarding the format, layout, or quality.
Remarks.
- (Task 4.1) Please describe a finite element model of a sandwich piezoelectric laminated beam according to Figure 1: length of beam, width of beam, and thickness of PZT layer. This information will be helpful for PhD students who study piezoelectric materials and in future increase your citations.
- Describe finite element model: SOLID5 or SOLID226 was used in the finite element calculation. Don’t forget to show the number of elements.
- Does not understand what the boundary conditions are in Table 4.
- Line 324. Did you calculate the natural frequencies for a sandwich beam considering the prestress effect resulting from temperature influence? Or did you use a different approach?
- Task 6 is not Patents.
Author Response

(The authors gave the same response as above.)

Round 2
Reviewer 3 Report
Comments and Suggestions for Authors
The work has been improved, as required by reviewers, and it is now suitable for publication at the present state.
Reviewer 5 Report
Comments and Suggestions for Authors
I think that the manuscript is perfectly readable and have no objections on that front.